# Charge-state lifetimes of single molecules on few monolayers of NaCl

Katharina Kaiser [1,3] ✉, Leonard-Alexander Lieske[1], Jascha Repp [2] & Leo Gross [1] ✉

In molecular tunnel junctions, where the molecule is decoupled from the electrodes by few-monolayers-thin insulating layers, resonant charge transport takes place by sequential charge transfer to and from the molecule which implies transient charging of the molecule. The corresponding charge state transitions, which involve tunneling through the insulating decoupling layers, are crucial for understanding electrically driven processes such as electroluminescence or photocurrent generation in such a geometry. Here, we use scanning tunneling microscopy to investigate the discharging of single ZnPc and $H_2$Pc molecules through NaCl films of 3 to 5 monolayers thickness on Cu(111) and Au(111). To this end, we approach the tip to the molecule at resonant tunnel conditions up to a regime where charge transport is limited by tunneling through the NaCl film. The resulting saturation of the tunnel current is a direct measure of the lifetimes of the anionic and cationic states, i.e., the molecule's charge-state lifetime, and thus provides a means to study charge dynamics and, thereby, exciton dynamics. Comparison of anion and cation lifetimes on different substrates reveals the critical role of the level alignment with the insulator's conduction and valence band, and the metal-insulator interface state.

Single-molecule charge transfer plays a significant role in many areas, from molecular electronics[1,2], single-molecule light emission[3–9], photocurrent generation[10] to natural processes such as photosynthesis[11]. Over the past 20 years, the tremendous progress in high-resolution scanning probe microscopy has facilitated the investigation of single molecules with charge state control[5,8,12–17]. In scanning tunneling microscopy (STM), this was enabled by introducing a thin insulating film as a decoupling layer between molecule and metallic substrate, preventing hybridization between molecule and substrate while still allowing charge transfer and preserving a sufficient conductance for STM[3,18–20]. This facilitates, for example, mapping molecular ion resonances, which is based on a transient change in the charge state of the molecule[19]. For the investigation of molecular electroluminescence in STM-induced luminescence (STML) experiments, the decoupling layer serves two purposes: It reduces luminescence quenching from the

metallic substrate[3,21–23], and, due to the finite lifetime of charged species, it enables an exciton formation mechanism based on subsequent charge transfer from tip and sample[7,24].

Such experiments on thin insulating films have in common that at sufficiently high bias voltages, sequential tunneling through a molecular resonance sets in. In this two-step sequential tunneling process, the molecule is transiently charged by a tunneling event between molecule and tip, followed by a tunneling event between molecule and metallic substrate. In almost all cases, the former tunneling event involving the tip is the current-limiting factor, such that little is known about the rate of the second tunneling transition involving the substrate. However, the latter can be critical for the interpretation of experimental results. For example, the aforementioned sequential tunneling process can—depending on the level alignment—lead to the formation of an excited state[3,7,8], which can subsequently decay under

[1]IBM Research Europe—Zurich, Säumerstrasse 4, 8803 Rüschlikon, Switzerland. [2]Department of Physics, University of Regensburg, Universitätsstraße 31, 93053 Regensburg, Germany. [3]Present address: Université de Strasbourg, CNRS, IPCMS, UMR 7504, F-67000 Strasbourg, France. ✉ e-mail: katharina.kaiser@ipcms.unistra.fr; LGR@zurich.ibm.com

the emission of a photon. The excitation mechanism is fundamentally different from optical excitation because it entails a two-step process[8,9,25], i.e., charging from the tip and subsequent charge transfer to the substrate. Hence, the entire cycle of electroluminescence including the emission of a photon already involves (at least) three transitions with their respective rates. In addition, the creation of the exciton by charge transfer competes with the neutralization of the molecule to its neutral ground state involving even a fourth rate. Thus, the charge-state lifetime of the adsorbed molecule needs to be taken into account in any consideration of dynamics in STML experiments, all-electric pump-probe measurements of excited states[24,26–30] as well as yields in photocurrent generation[10].

The average elapsed time between charging by tunneling between tip and molecule, and neutralization by charge transfer between molecule and substrate depends on the tunneling probability between molecule and metallic substrate and thus on the thickness of the insulating film[15,16,31,32]. Although this time is a property of the entire system and occurs in an out-of-equilibrium situation, in the following, we refer to this quantity as the charge-state lifetime.

One way of experimentally determining charge-state lifetimes has been demonstrated for Cl-vacancies in NaCl films of various thicknesses[31]. Analogously to surface-adsorbed molecules, these defects exhibit electronic resonances and can be transiently charged at sufficiently high bias voltages. At resonance, the system represents a double-barrier tunnel junction with one barrier corresponding to tunneling between tip and defect (vacuum barrier) and the other corresponding to tunneling between defect and metal substrate (NaCl barrier). At tunnel conditions in typical STM experiments, the charging step by tunneling through vacuum is the rate-limiting step and therefore determines the measured current $I$. In this (usual) regime, the current $I(z)$ increases exponentially with decreasing tip height $z$. At close distances, however, the time for charging the neutral defect by tunneling through vacuum ($\tau_c$) can become smaller than the time to discharge the defect through the insulating film (i.e., the charge-state lifetime $\tau_d$), and thus, $I(z)$ reaches a $z$-independent saturation current $I_{sat}$ for small $z$. Specifically, at the first (positive and negative) ion resonance, the doubly-charged state is energetically not available due to Coulomb repulsion, such that in this regime, $I$ is limited by the defect's charge-state lifetime $\tau_d$ and for small $z$ no longer depends on $z$[22,29–32]. Hence, under these conditions, one can directly deduce $\tau_d = qI_{sat}^{-1}$, with the elementary charge $q$, from the saturation current $I_{sat}$. Since Cl-vacancy states are localized within the top-most NaCl layer, possess an $s$-wave character as well as strong lateral confinement, and have no occupied state in the relevant energy range, it is not straightforward to generalize the results from vacancies to molecules.

Here, by extending this approach to single molecules, as sketched in Fig. 1, we investigated charge-state lifetimes of ZnPc and H$_2$Pc molecules, both of which are frequently used model systems in STML experiments[4,29,33], adsorbed on 3–5 monolayers (ML) of NaCl on Au(111), i.e., NaCl/Au(111) and Cu, i.e., NaCl/Cu(111). Comparing results on surfaces of different work functions, Au(111) and Cu(111), allows us to shift the molecular electronic states with respect to the sample's electrochemical potential in order to probe the effect of different possible tunnel channels for the neutralization event.

## Results

### Saturated tunnel current

We performed current-versus-distance spectroscopy $I(z)$ within the electronic resonances of ZnPc and H$_2$Pc molecules adsorbed on 3 to 5 ML thick NaCl films on Cu(111) and Au(111) substrates. To that end, we applied voltages $V$ at and, in absolute values, up to few 100 mV above the respective peak positions in d$I$/d$V$, $V_{PIR}$ and $V_{NIR}$ (see Methods as well as Fig. 2e, f and Supplementary Fig. 2 for scanning tunneling spectra and additional information). Figure 2a, b shows $I(z)$-curves

recorded above a lobe of a ZnPc at the first negative ion resonance (NIR) and positive ion resonance (PIR), respectively, for different NaCl film thicknesses $d_{NaCl}$.

As in the case of Cl-vacancies, the spectra in Fig. 2a, b show two distinctly different regimes: At large $z$, the $I(z)$ spectra exhibit an exponential $z$-dependence; the current-limiting tunnel process is the tunneling through vacuum between tip and molecule. For small $z$, $I(z)$ shows a plateau at the saturation current $I_{sat}$. This shows that at the electronic resonances tunneling between tip and metal substrate is governed by a two-step tunnel process via the molecule, whereas the contribution of direct tunneling, which should increase exponentially with decreasing $z$, is negligible. In addition, in the regime of saturation the tunnel current is governed by tunneling through the NaCl barrier, and the molecule is charged most of the time.

The lateral dependence of the current can be used to experimentally verify if the saturation observed in $I(z)$ is due to the effect described above. If the current was indeed limited by molecule-to-substrate tunneling, it should be also largely independent of the lateral tip position. Only if the tip was placed laterally off of the entire

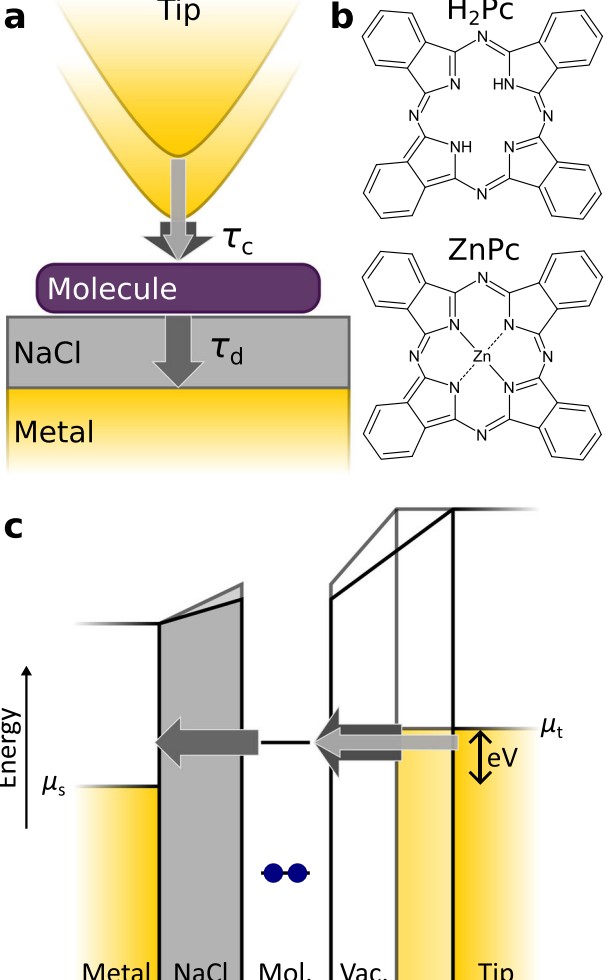

**Fig. 1 | Resonant tunneling through a molecule adsorbed on a thin NaCl film. a** Schematic depiction of the transient negative charging of the molecule at positive bias. Chemical structures of the utilized molecules (H$_2$Pc and ZnPc) are shown in **b**. The charging time through the vacuum barrier $\tau_c$ can be varied by changing the distance between tip and molecule; thicker arrows indicate smaller charging times. The discharging time, i.e., the charge-state lifetime, through the NaCl film $\tau_d$ is governed by the film thickness. The two cases that are schematically depicted here correspond to tip-molecule distances where $\tau_c$ is longer (long, thin arrow) and shorter (short, thick arrow) compared to $\tau_d$. **c** Corresponding one-electron picture of the double-barrier tunnel junction shown in **a**.

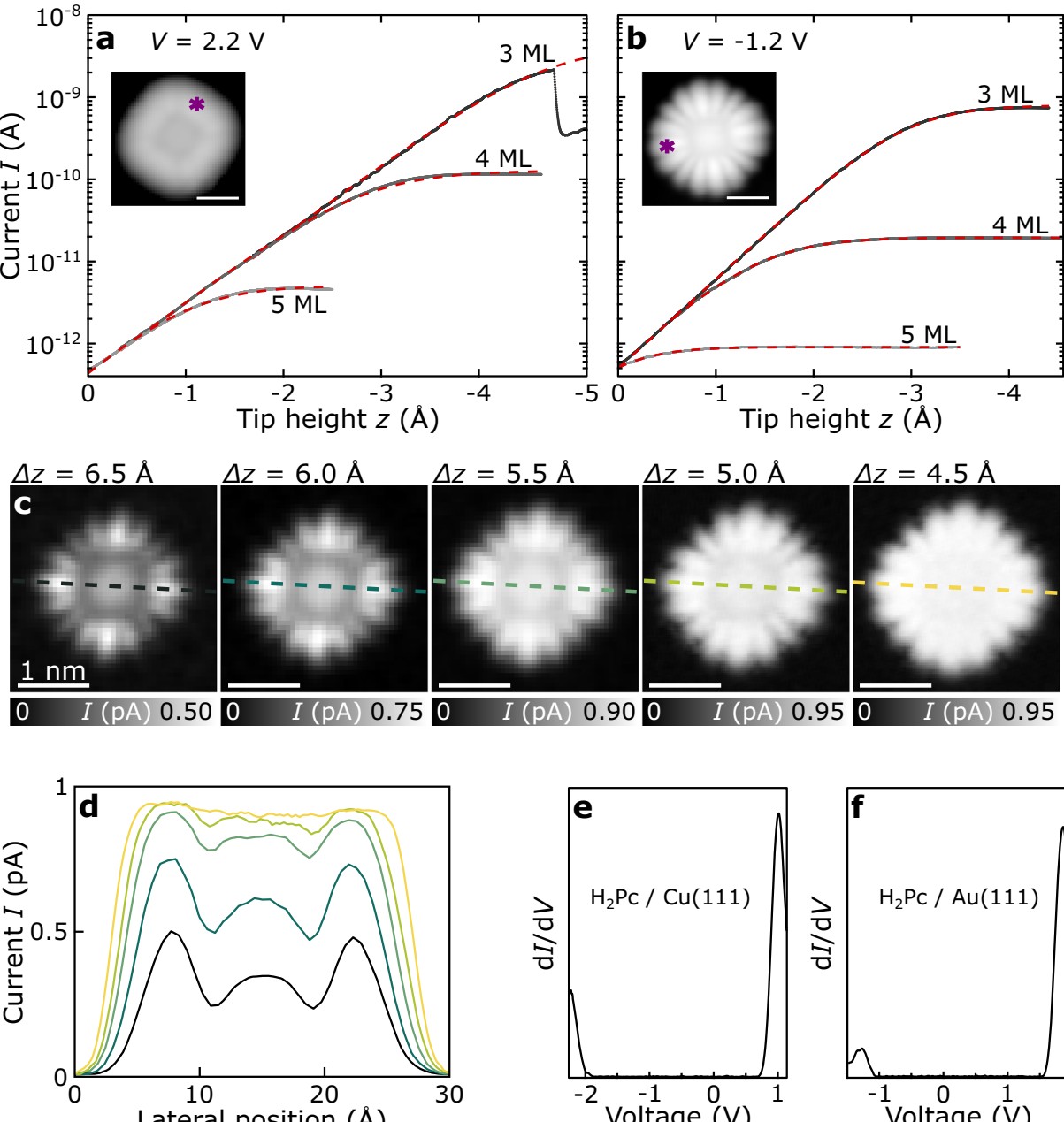

**Fig. 2 | Current as a function of tip height for ZnPc adsorbed on NaCl films of different thicknesses on Au(111).** I(z) spectroscopy at the (**a**) negative ion resonance (NIR) and (**b**) positive ion resonance (PIR) of ZnPc molecules adsorbed on 3–5 monolayer (ML) NaCl/Au(111). Fits to the data are shown in red. On 3 ML at NIR, the molecule dislocated at $z \approx -4.8$ Å, resulting in an abrupt change in current. The insets in **a** and **b** exemplify the lateral position of the tip, atop regions of high orbital density, during the spectra. **c** Constant-height STM images (V = –1.2 V) of ZnPc adsorbed on 5 ML NaCl/Au(111) recorded at different tip-sample distances. The tip-height offset $\Delta z$ is given with respect to the STM setpoint of V = –1.2 V, I = 0.5 pA above the bare NaCl surface. The scale bar corresponds to 1 nm and applies to all images in **c**. **d** Line profiles along the dashed colored lines indicated in **c**. **e**, **f** dI/dV as a function of V of $H_2Pc$ adsorbed on 4 ML NaCl on Cu(111) (setpoint V = 1.6 V, I = 0.5 pA, $\Delta z$ = –1 Å) **e** and 3 ML NaCl on Au(111) (setpoint V = 2 V, I = 0.5 pA, $\Delta z$ = –1 Å); obtained by numerical differentiation of constant-height I(V)-curves.

molecule, the current should drop. Constant-height STM images recorded at resonance, shown in Fig. 2c for ZnPc adsorbed on 5 ML NaCl recorded at the PIR, i.e., V = –1.2 V, show a transition to this regime. For a larger tip-height offset of $\Delta z$ = 6.5 Å, the constant-height STM map shows a maximum current of about 0.5 pA, which is smaller than the saturation current $I_{sat}$, and a significant variation of the tunnel current is observed as a function of the lateral tip position above the molecule, as usual[19].

Upon decreasing z, the current increases in all regions above the molecule, however, only until reaching the saturation current $I_{sat}$ for ZnPc/NaCl(5 ML)/Au(111) of about 0.9 pA, compare with Fig. 2b. At $\Delta z$ = 4.5 Å, we obtain a "flat-top" contrast with the saturation current reached at all positions above the molecule, and nodal planes can no longer be observed. This confirms that $\tau_d$ is independent of the position at which the charge had been attached to the molecule. Such current behavior in constant-height STM maps is reproduced for different film thicknesses and resonances, see Supplementary Fig. 1. Bias voltage dependent measurements reveal that $\tau_d$ remains the same within a few 100 mV around the peak voltage of the resonance and several 100 mV above it (see Supplementary Fig. 3a, b). By changing the total applied bias voltage (and tip height), the voltage drop across NaCl also changes by a small amount[34]. $I_{sat}$ being independent of

$V$ (for voltages associated to a certain ion resonance, see Supplementary Fig. 3a, b) and of the tip height $z$ (Fig. 2a, b) indicates that the charge-state lifetime does not vary significantly as a function of the voltage drop across NaCl for a specific ion resonance. A more detailed discussion of the voltage dependence and the partial voltage drops in vacuum and NaCl is given in the Supplementary Note 2.

## Determination of effective barrier heights

The $I(z)$-curves shown in Fig. 2a, b can be fitted assuming a total rate $\Gamma_{tot}$ for the entire charge transfer process between tip and metal substrate that comprises the charging time $\tau_c$ of the neutral molecule by tunneling through the vacuum barrier, with an exponential $z$-dependence, and the charge-state lifetime $\tau_d$ by tunneling through the NaCl film that is independent of $z$.

$$\frac{1}{\Gamma_{tot}}(z) = \tau_c(z) + \tau_d = \frac{1}{\Gamma_{vac}(z)} + \tau_d = \frac{1}{\Gamma_{vac}^0 \exp(-2\kappa_{vac} \cdot z)} + \tau_d \quad (1)$$

$$I(z) = q \cdot \Gamma_{tot} = \frac{\Gamma_{vac}^0 \cdot q}{\exp(2\kappa_{vac} \cdot z) + \Gamma_{vac}^0 \cdot \tau_d} \quad (2)$$

Here, $\Gamma_{vac}(z)$ is the rate at which the neutral molecule is charged via the tip through the vacuum barrier (as a function of $z$) and $\Gamma_{vac}^0$ is that rate for $z = 0$. $\Gamma_{vac}^0$ as well as $\kappa_{vac} = \frac{\sqrt{2m_e\phi_{vac}}}{\hbar}$ are fitting parameters, with the electron mass $m_e$ and the effective vacuum barrier height (between molecule and tip) $\Phi_{vac}$. The fits to the experimental $I(z)$-curves are shown as dashed red lines in Fig. 2a, b, respectively.

Figure 3 shows the extracted saturation currents and related charge-state lifetimes $\tau_d$ of anions and cations of ZnPc and H$_2$Pc on NaCl films on Au(111) and Cu(111) as a function of $d_{NaCl}$. The charge-state lifetimes $\tau_d$ range from around 50 ps (3 ML NaCl/Cu(111)) to 20 ns (5 ML NaCl/Au(111)). The previously reported charge-state lifetimes of the anionic charge states of Cl-vacancies in NaCl[31] are about one order of magnitude smaller (on the order of 1 ns for 5 ML NaCl) compared to the charge-state lifetimes of the investigated negatively charged molecules on 5 ML NaCl. Aside from other effects, e.g., different barrier

heights and energetic positions of the resonances, this relates to the probed Cl-vacancies being located within the top monolayer of the NaCl-film, whereas molecules are adsorbed on top of the NaCl film with an adsorption height of approximately 0.3 nm[35].

As conventional $I(z)$ spectroscopy can be used to extract $\Phi_{vac}$[36,37], the slope $m = \ln(I_{sat})d_{NaCl}^{-1}$ provides access to the effective NaCl barrier height $\Phi_{NaCl}$. Comparing the data in Fig. 3, we make two important observations: First, the slopes $m$ for both molecules and at both ion resonances (PIR and NIR) are very similar, with $m \approx -2.9$ ML$^{-1}$ for Cu(111), see Fig. 3a, and m $\approx -3.4$ ML$^{-1}$ for Au(111), see Fig. 3b. That is, for both cations and anions, $I_{sat}$ decreases ($\tau_d$ increases) by a factor of 18 for Cu(111) and by a factor of 30 for Au(111), per added monolayer NaCl. The corresponding extracted effective NaCl barrier-heights are $\Phi_{NaCl/Cu} \approx 0.9$ eV on Cu(111) and $\Phi_{NaCl/Au} \approx 1.3$ eV on Au(111). The full list of slopes and effective barrier heights is reported in the supplementary information in Supplementary Table 1. Second, while on Cu(111), the lifetimes of cations (PIR) and anions (NIR) are similar (see Fig. 3a), on Au(111), the lifetime of the cations is about one order of magnitude longer compared to that of the anions (NIR) (comparing same NaCl layer thicknesses, see Fig. 3b).

The first observation, namely the similar $m$ for positive and negative ion resonance, indicates that the NaCl effective barrier heights $\Phi_{NaCl}$ for the transitions from anion and cation to the neutral state are similar. This is different compared to tunneling through vacuum, where the barrier height increases with increasing energy difference to the vacuum level and therefore is smaller for the NIR compared to the PIR.

## What affects the charge-state lifetimes?

In the following, we will discuss these observations based on the energetic alignment of potential tunnel channels that may be involved in the neutralization process with respect to the substrate states (Fig. 4). Note that the picture presented in Fig. 4 is a sketch with estimations based on previous experiments (see Supplementary Fig. 5), which could be refined and better quantified by future experiments and theoretical investigations.

In general, neutralization of a charged species in a $D_0^{+/-}$ state can, depending on the level alignment between molecule and metal

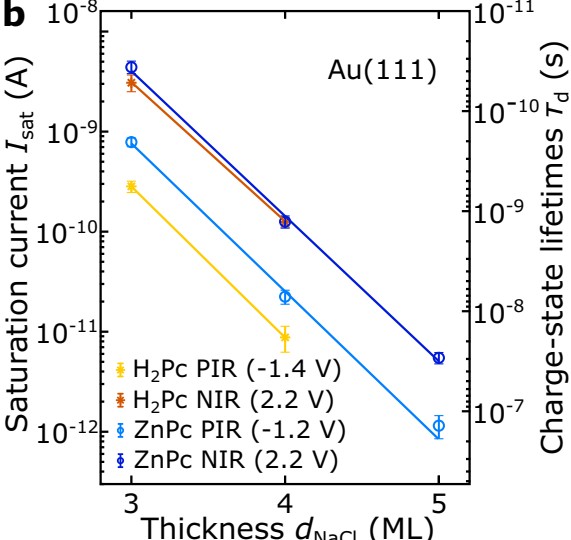

**Fig. 3 | Charge-state lifetimes.** Measured saturation currents I$_{sat}$ (left axis, log scale) and corresponding charge-state lifetimes τ$_d$ (right axis, inverted log scale) of ZnPc (light and dark blue symbols) and H$_2$Pc (yellow and orange symbols) on NaCl on (**a**) Cu(111) and (**b**) Au(111) as a function of NaCl thickness. The saturation current was recorded at voltages within the ion resonances. The voltages that were typically used to record the I(z)-curves are indicated in the legends. Their absolute values are slightly larger than the voltages at which the corresponding resonance is seen in dI/dV. The solid lines are linear fits to the data. For detailed information on the statistical significance of the error bars, see Supplementary Table 2.

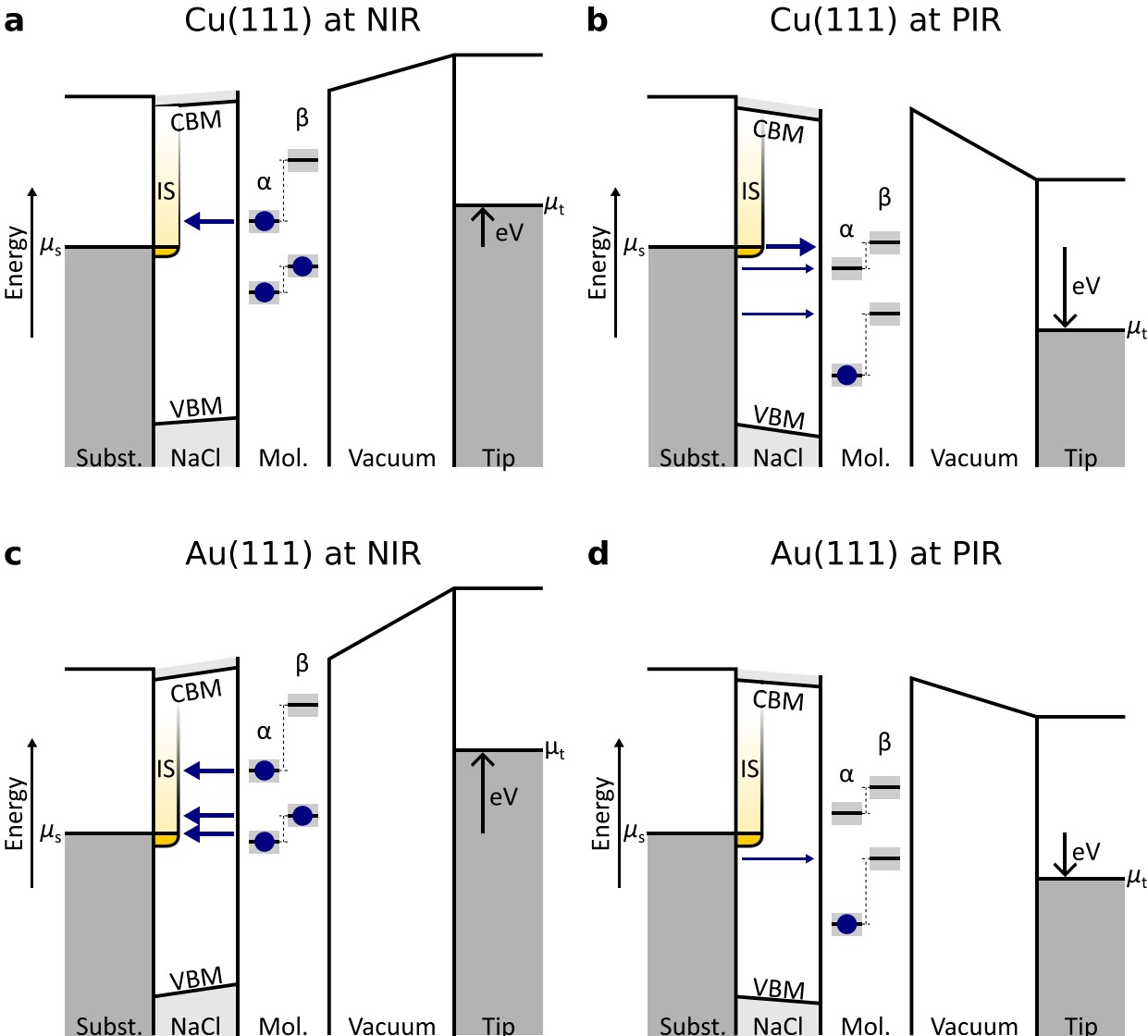

**Fig. 4 | Transition from a molecule's transiently charged state to the neutral charge state by tunneling through the NaCl barrier.** The frontier molecular levels, chemical potentials of tip ($\mu_t$) and sample ($\mu_s$), and the interface state (IS) are indicated. The shown level alignment corresponds to voltages of the ion resonances deduced from the positions of the neutral molecule's ion resonances and STML. The dashed lines indicate which single electron states derive from the HOMO (lower pair of states) and LUMO (higher pair of states) of the neutral molecule, the corresponding α and β spin channels are indicated. The gray shaded area indicates the linewidth of the levels. Because of different work functions, the vacuum-level aligned molecular ion resonances are shifted to larger bias values on Au(111) compared to Cu(111)[12,32]. Thicker (thinner) arrows indicate channels that involve (do not involve) the IS. In **a**–**c**, the IS contributes to at least one channel, while in **d**, it does not. A more detailed depiction of how the shown level alignment was derived is shown in the Supplementary Fig. 5. By schematically showing the conduction band minimum (CBM) and valence band maximum (VBM) of NaCl we indicate the need to consider the band structure of NaCl for tunneling. Refs. [67–69]. suggest that the CBM is roughly aligned with the vacuum level while the band gap is at least 8 eV (see Supplementary Note 5).

substrate, proceed via several channels, i.e., in addition to transitions to the neutral ground state ($S_0$) transitions to excited states (e.g., $T_1$, $S_1$) could be energetically allowed[3,4,9,38]. Because only a small fraction of the applied bias drops in the NaCl film, about 10–20% for the used geometries (see, e.g., ref. [15], and Supplementary Note 2), by changing the applied bias voltage we can shift the molecular levels only within a very limited energy window with respect to the substrate's electrochemical potential. Instead, we can use metal substrates with different work functions to compare situations where only the channel to the neutral ground state is energetically allowed, to situations where also channels to excited states are possible. Although the exact positions of the anion's/cation's electronic levels are not accessible in STM on ultrathin insulating films, we can deduce for which cases excited states can be formed in the neutralization step, e.g., from comparison to STML measurements and comparing the energies of the PIR and NIR

with excited state energies (i.e., energies of optical transitions). These information allow us to draw an estimated picture of the level alignments between the transiently charged molecules and the substrate. We assume that on Cu(111) the anion, formed at the NIR at about $V_{NIR} = 1$ V, only neutralizes to the ground state $S_0$, whereas the cation, formed at the PIR at about $V_{PIR} = -2.2$ V, can also form the excited $S_1$ and $T_1$ state upon neutralization. On the other side, we assume that on Au(111) the cation, formed at the PIR at about $V_{PIR} = -1.1$ V, only neutralizes to $S_0$, whereas the anion, formed at the NIR at about $V_{NIR} = 2.1$ V, can also form the excited $S_1$ and $T_1$ state[5,38]. For a more detailed discussion see the Supplementary Note 3 and Supplementary Fig. 5. This energetic alignment of the different channels is considered and reflected in the schematics shown in Fig. 4, and it allows us to discuss how the lifetime changes when different channels are open for neutralization. Note that the level alignments in Fig. 4 depict the

molecular resonances after the charging event via the tip, which differ from those of the neutral molecule (Fig. 2e, f) due to, e.g., Coulomb repulsion, lifted spin degeneracy and reorganization energy[7,15].

The experimentally observed similar $m$ for anion and cation indicates similar effective barrier heights for the charged-to-neutral transitions. If here the effective barrier height simply corresponded to the respective energetic difference to the vacuum level (or another single fixed energy level, e.g., the conduction band minimum CBM or valence band maximum VBM), the extracted effective barrier height $\Phi_{NaCl}$ would be significantly different for the PIR (neutralization of the cation) compared to the NIR (neutralization of the anion). Note that in general, for the NIR (PIR) neutralization occurs by electrons tunneling at energies above (below) the electrochemical potential of the metal sample, i.e., to unoccupied (from occupied) sample states. We hypothesize that the similar $m$ result from the effective NaCl barrier height $\Phi_{NaCl}$ being determined by the relative energetic position of the tunnel channels with respect to the band structure of the NaCl film. It has been shown that for tunnel processes involving states that are energetically near the VBM, hole tunneling, for which the tunnel barrier height is given with respect to the VBM and thus decreases with decreasing carrier energy, can dominate[39]. Especially for the cation (Fig. 4b, d), the energetic difference of one of the tunnel channels to the VBM becomes comparable to the energetic difference of tunnel channels to the CBM and vacuum level in other cases, which could explain its similarly small $m$ compared to the anion (Fig. 4a, c). However, since the different neutralization processes involve tunnel channels with different energetic alignment with respect to the substrate's states and thus with respect to CBM and VBM (see Fig. 4), it seems unlikely that the effective barrier heights result from the energetic separation of these channels to either VBM or CBM alone. Instead, the similar $m$ suggest that the effective barrier height for the neutralization of the ionic molecules is nearly energy independent for the different processes studied in this work. In fact, tunneling through a solid is correctly described by a complex band structure[40–42]: inside the band gap the wave vectors become imaginary and the wave functions decay exponentially into the bulk. Such imaginary wave vectors adequately describe the tunneling and thereby translate into an effective tunneling barrier. Complex band structure calculations for the wide-band-gap insulator MgO suggest that deep inside the band gap the imaginary wave vector becomes nearly independent of energy[40–42]. We expect the complex band structure of NaCl to be qualitatively similar, which would explain our observation of $m$ being nearly independent of energy. Note that, in direct vicinity to CBM or VBM, we expect the effective barrier height to be smaller and exhibit a stronger energy dependence compared to deep in the band gap, as observed for doubly charged Cl vacancies[31]. The potential landscape being influenced by the different transient charge states of the molecule as the initial state of the tunneling process through NaCl may further influence the effective barrier heights.

Since we find similar slopes $m$ in $\ln(I_{sat})$ for the decharging of anion and cation from the sample, the second observation, i.e., the cation lifetimes on Au(111) being about one order of magnitude longer than that of the anions, cannot be explained by differences in the effective barrier height for the involved tunneling events. Instead, the observed differences can be rationalized by considering the sample's local density of states (LDOS) at the energy of the ion resonances. The interface state (IS) band that descends from the Shockley surface state of noble metal (111) surfaces[43,44] upon adsorption of a dielectric[45–48] extends the sample's LDOS, and thus reduces the length of the tunnel path through NaCl for transitions including the IS, and modulates it[34,46,49,50]. The onset of the NaCl/metal IS for Cu(111) is at about $V = -220$ mV and for Au(111) at $V = -270$ mV[45,46]. Figure 4 shows schematically in a single-electron picture how the IS potentially contributes to tunneling through the NaCl barrier: It contributes for all systems studied, except for the cation on Au(111), where presumably only

tunneling from the substrate at energies below the onset of the IS can neutralize the cation, see Fig. 4d. We propose that this is the main reason for the comparably long lifetimes of the cations on Au(111). In contrast, for the cations on Cu(111), tunneling from the substrate to neutral excited states is energetically possible[5,7,8,38], opening channels at IS energies, see Fig. 4b. In addition, the number of accessible tunnel channels might influence the lifetime. Differences in the spatial extension of the singly occupied orbitals' wavefunctions will also affect the relative contributions of the different channels. The latter argument is not considered in Fig. 4.

In summary, on both surfaces, the neutralization of the anion and cation by tunneling through the NaCl film exhibits very similar effective barrier heights, although the energetic separation of the involved tunnel channels with respect to the vacuum level, the VBM, and the CBM is very different. This leads us to the conclusion that $\Phi_{NaCl}$ results from the relative energetic position of the tunnel channels inside the band gap of the NaCl film. Deep inside the band gap, $\Phi_{NaCl}$ does not seem to vary significantly with the relative energetic position of the tunnel channels. In addition, on Au(111), the longer lifetimes observed at the PIR compared to the NIR – despite similar effective barrier heights – indicate tunneling across different distances. This can be rationalized by the increased LDOS due to the IS, that contributes to at least one channel in all situations (Fig. 4a–c) except for the PIR on Au(111) (Fig. 4d).

As a side remark, if several channels are open, i.e., as for the PIR on Cu(111), see Fig. 4b, and for the NIR on Au(111), see Fig. 4c, the fastest channels should dominantly govern the charge-state lifetime and the measured effective barrier height. However, in our experiment, we cannot separately measure the rates of competing channels. Based on our results and arguments discussed above we would expect that the channels that involve the IS, sketched bold in Fig. 4, are faster and thus dominant. STML showing light emission from the $S_1$ state for the NIR on Au(111)[5,17,38], indicates that out of the three channels that involve the IS, shown in Fig. 4c, the channel that is lowest in energy contributes at least significantly.

We note that momentum conservation as well as the wave vector parallel to the surface ($\mathbf{k}_\parallel$) can additionally influence the tunneling probabilities[51–54], but estimating this influence based on the momenta of the involved states[46,55,56] is beyond the scope of this work.

## Excited and higher charge states

At voltages sufficiently exceeding the first electronic resonances, additional tunnel channels can be accessed[31,57,58]. Figure 5a shows $I(z)$ spectroscopy of ZnPc on NaCl(5 ML)/Au(111) at different negative sample voltages. Very similar $I(z)$ spectra and saturation currents $I_{sat}$ are measured for $V$ from $-1.6$ V to $-2.4$ V. At $V = -2.5$ V, however, the current shows a much less pronounced plateau at about $z = -0.5$ Å and then further increases with decreasing $z$. For $z < -1.5$ Å, the current significantly exceeds the saturation current of $I_{sat} \approx 0.9$ pA measured at less negative voltages. This is also visualized in constant-height STM images at $V = -2.5$ V at different tip-sample distances (Fig. 5b). For $\Delta z > 5.0$ Å, the same behavior as for smaller negative voltages is observed, i.e., with decreasing $z$, the current reaches saturation at an increasing number of lateral positions above the molecule until a flat-top current image is observed (at $\Delta z = 5.0$ Å). Going closer with the tip ($\Delta z < 4$ Å), we observe that the current increases further at some lateral positions but remains "flat" in other regions above the molecule. Interestingly, the spatial distribution of regions of increased current shows a ring of 12 maxima, as on a clock, with bright maxima at positions 3, 6, 9, 12 o'clock and two equally spaced fainter maxima in between. This corresponds to the shape and symmetry observed for the lowest unoccupied molecular orbital (LUMO) rather than the highest occupied molecular orbital (HOMO) density, which exhibits three (not two) equally spaced fainter lobes between the bright lobes (see Fig. 5d for comparison).

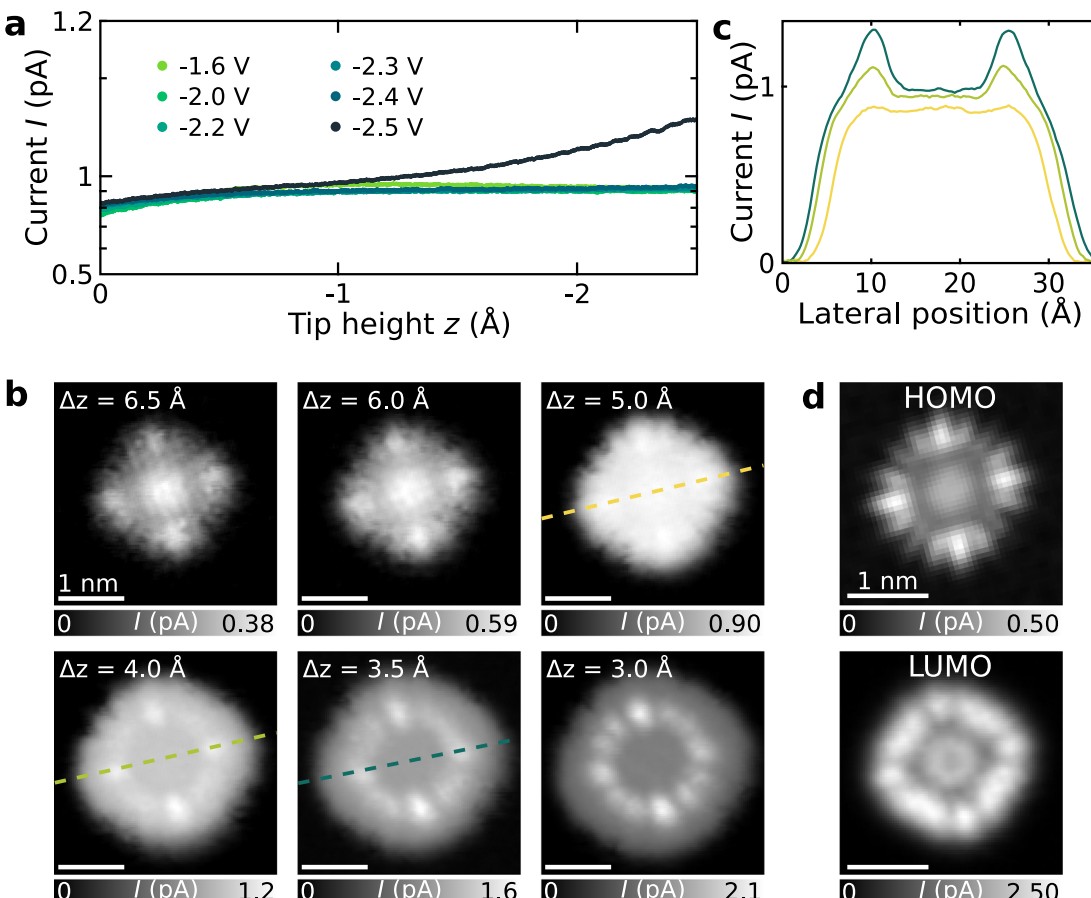

**Fig. 5 | Contribution of higher lying states of ZnPc on 5 ML NaCl/Au(111). a** I(z)-curves atop one of the lobes of the ZnPc HOMO at different V from −1.6 to −2.5 V. **b** Constant-height STM images on ZnPc adsorbed on 5 ML NaCl/Au(111) at different Δz recorded at V = −2.5 V. The colored dashed lines indicate the position of the corresponding line profiles shown in **c**. The scale bar corresponds to 1 nm and applies to all images. Δz is given with respect to the STM setpoint of V = −2.5 V, I = 0.5 pA above the bare NaCl surface. **d** Constant-height STM images of ZnPc adsorbed on 5 ML NaCl/Au(111) at PIR (HOMO) and NIR (LUMO) for comparison.

The contrast that we observe at V = −2.5 V for small z (at Δz = 3 Å) could be explained by transitions involving higher-lying states of ZnPc, such as for example higher charge states[57–59], i.e., the dication ($S_0^{2+}$), or trionic states ($D_n^+$)[5,17]. Some potentially accessible states and corresponding transitions are shown in a many-body energy diagram in Supplementary Fig. 3.

The dication, i.e., the doubly positively charged ground state ($S_0^{2+}$), could be accessed by applying sufficiently large currents and bias voltages in a two-electron process, which becomes significant in a regime where the $D_0^+ \rightarrow S_0$ transition (neutralization by tunneling through NaCl) is slower than $D_0^+ \rightarrow S_0^{2+}$ (tunneling another electron to the tip), i.e., at small z. We observe a corresponding behavior in the I(z)-curves in Fig. 5a and the constant-height STM maps in Fig. 5b, where for Δz = 5 Å, we observe the saturation of the tunnel channel corresponding to the $S_0 \rightarrow D_0^+$ transition as well as a subsequent increase of the current at Δz < 5 Å.

Due to a non-zero population of $S_0^{2+}$, transitions into other higher-lying states, such as the cation's excited state $D_1^+$, become accessible. Thus, different tunnel channels can contribute to the overall tunnel current[60,61], and the contrast in STM results from a superposition of these channels[60,62]. In the transition $D_1^+ \rightarrow S_0^{2+}$, for example, an electron is removed from the former LUMO (which is singly occupied in $D_1^+$) by tunneling through the vacuum barrier, which could explain the observed contrast in Fig. 5b at Δz = 3 Å.

The results in Fig. 5 demonstrate that additional transitions can contribute to the overall tunnel current when increasing the bias voltage, and their relative contribution can be tuned with tip-sample distance. These transitions can play an important role, for example, in the formation of excited states in STML experiments, where exciton formation via the singly charged molecule is not always energetically possible and is thus sometimes only observed at higher-lying ion resonances[5,27,63–65].

In conclusion, we reported lifetimes of transiently charged molecules on thin NaCl films on Au(111) and Cu(111) surfaces. Previously, Hanbury Brown-Twiss interferometry and phase fluorometry in combination with STML have been used to access exciton dynamics in single molecules adsorbed on ultrathin insulating films[27,29]. The extracted lifetimes comprise time constants for both excitation and decay of molecular excitons and are of similar magnitude as the charge-state lifetimes reported here (on the order of 500 ps for 3 ML NaCl). This indicates that, for certain geometries, the neutralization process from the substrate is the rate limiting process in the formation and decay of molecular excitons and, thus, the previously extracted time constants are likely dominated by the charge-state lifetime. Our results further indicate that the effective NaCl barrier heights for neutralization are governed by the energetic alignment of the tunnel channels inside the band gap of the insulating decoupling layer. Moreover, the energetic alignment of the tunnel channels for neutralization with the interface state significantly impacts the charge-state lifetime. Our results provide an improved understanding of the tunnel processes in these relevant double-barrier tunnel junctions and a quantification of the lifetime of transiently charged molecules, important for understanding excited-state formation by charge attachment in the growing field of STML experiments.

## Methods

We performed the experiments in a home-built low-temperature ($T \approx 5$ K) combined STM/AFM system operated under UHV conditions and at a base pressure of $1 \times 10^{-10}$ mbar. The voltage $V$ was applied to the sample. The metal substrates were cleaned by repeated Ne$^+$ ion sputtering and annealing cycles. NaCl was deposited at sample temperatures between 250 K and 300 K[45]. ZnPc and H$_2$Pc (Sigma-Aldrich, purity >97%) were sublimed onto the cold ($T \approx 10$ K) substrate from a Si-wafer by resistive heating.

For constant-height STM and $I(z)$ spectroscopy, we approach the tip by the tip-height offset $\Delta z$ from a given STM-controlled setpoint, indicated in each caption. In $I(z)$ spectroscopy the offset for the tip height $z$ is chosen such that at $I = 0.5$ pA, $z$ is 0 Å in every spectrum. Increases in $z$ and $\Delta z$ correspond to increases in tip-sample distance. We define the voltages corresponding to the peaks in d$I$/d$V$ as $V_{PIR}$ and $V_{NIR}$, respectively. The voltages $V$ used in the experiment for probing the lifetime at PIR and NIR were chosen such that, by varying $z$ and thus changing the lever arm for the voltage drop across NaCl, the molecular electronic resonances stay accessible. To this end, we typically used for all tip heights slightly larger voltages (up to few 100 meV) compared to the respective peak positions in d$I$/d$V$ for the larger tip heights probed.

The effective barrier heights for tunneling through NaCl were determined from the exponent of the saturation current as a function of NaCl thickness, $I_{sat}(d_{NaCl})$, i.e., the slope of $\ln(I_{sat}(d_{NaCl}))$, according to Eq. (1) in the Supplementary Information.

### Reporting summary

Further information on research design is available in the Nature Portfolio Reporting Summary linked to this article.

## Data availability

All data supporting the findings of this study are available from the corresponding authors upon request. The $I(z)$- and $I(V)$-spectra as well as the STM images presented in the Figs. 1, 2, and 5 and the Supplementary Figs. 1–3, and 6 are available via Zenodo[66].

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

## Acknowledgements

We thank Rolf Allenspach, Guillaume Schull, Anna Rosławska, Song Jiang, Kirill Vasilev, Daniel Wegner, Florian Albrecht, Shadi Fatayer, Fabian Paschke, Armin Knoll, Shantanu Mishra, and Stefan Fölsch for discussions and comments. This work was supported by the ERC Synergy Grant MolDAM (no. 951519, L.G. and J.R.), the EU FET-OPEN project SPRING (no. 863098, L.G.), and the H2020-MSCA-ITN ULTIMATE (no. 813036, L.G.).

## Author contributions

L.G. and K.K. designed the experiment. K.K., L.-A.L., and L.G. performed the experiments. K.K., L.G., L.-A.L., and J.R. discussed the results and wrote the manuscript.

## Competing interests

The authors declare no competing interests.
