## [Peer Review File · Nature Communications]

nature portfolio

Peer Review FileReviewer comments, first round

Reviewer #1 (Remarks to the Author):

The work "Charge-state lifetime of single molecules on ultrathin insulating films" investigates transport through phthalocyanine molecules deposited on 3-5ML of NaCl on Au111 and Cu111. It brings compelling evidence that the lifetimes of charged molecular states excited through the resonant transport channels of the system depend fundamentally on the conductivity across the NaCl spacer and consequently govern formation of excited states in the chromophore. The authors put the results into perspective with previous results on Cl-vacancies and evaluate the tunnelling barriers and tunnelling rates (lifetimes) quantitatively.

This is an interesting study that has the potential to shed some light on the recently discussed topics in the STML community. It is a surprisingly solid result considering the simple method employed for this study. However, the discussion is very dense and certain points seem quite confusing. I would recommend a thorough revision of the interpretation and possibly adding more data, before entering the second round. Here are my comments and questions that may need to be reflected in the manuscript.

1) Ballistic transport of electrons is not considered in the analysis. It may be obvious to the authors, but which are the grounds to disregard it?

2) Band models in Fig.1a and in Fig.4 raise these concerns:

i) As drawn in these schemes, voltage drop seems to be constant, regardless of the bias and vacuum barrier width. This is commented in the part 2 of the SI, and stated that upon reducing the vacuum barrier, the change is expected to be small. At the same time, the total barrier height is *assumed* to be 10-20% at NaCl, citing DOI:10.1038/s41565-018-0087-1 (which puts the value at 17% for 14 ML of NaCl, which seems rather inconsistent with the current manuscript). The problem here is that there is no convincing data for this quite a striking assumption of a constant voltage drop.

In particular, positions of NIR and PIR are not documented by any STS measurements in this work. It would be a good idea to add them to the SI. It can (dis)prove the independence on the z-distance and layer thickness, as it has been done in DOI:10.1103/PhysRevB.98.201403.

ii) Upon charge transfer between an electrode and an orbital, and following exciton, energy level renormalization should occur, but it is not considered and not discussed at all. Instead the shift of the PIR and NIR is interpreted solely as a result of the voltage drop. This contradicts some previous works where the role of the intramolecular Coulomb interaction is suggested as the main reason e.g. in DOI:10.1021/acs.nanolett.8b04484

iii) The role of the interface state is probably not needed to explain the lower saturation currents for Au(111) PIR, because the tunnelling barrier alone for the corresponding electron transfer is the highest among all the considered cases.

Remarks to consider:

Shouldn't the slope equation be correctly: $m = \ln(1/I_{\text{sat}})/d_{\text{NaCl}}$?

While the title, abstract and introduction mention charge-state lifetimes, the τ_d is named as "discharging time" near to the main equation. It could be better to name it consistently throughout the paper.

It will be very interesting to put the charge-state lifetime into context with the past STML measurements that measured "lifetimes", e.g. DOI:10.1038/s41467-017-00681-7,

DOI: 10.1021/acsnano.1c01318.

Fig.S3 depicts the scheme of possible states and transitions between them. In the light of recent discussions in the community, the position of the states at the 'Free energy' axis could be set quite precisely according to the onsets of the PIR and NIR energies. Also, in the many-body picture, direct transition $S0 \rightarrow D1+$ is possible upon charge transfer between the tip and the molecule.

Reviewer #2 (Remarks to the Author):

The manuscript "Charge-state lifetimes of single molecules on ultrathin insulating films" presents the lifetime measurement of the positive- and negative-ion resonance states of ZnPc and H2Pc molecules adsorbed on Au(111) and Cu(111) and rationalizes the dependence of the measured lifetimes on the thickness of the NaCl decoupling layer. As the authors mention, the experimental STM technique used for the measurements is not new and was already applied by one of the co-authors to determine the lifetimes of Cl vacancies on NaCl overlayers of varying thicknesses. Trying to defend the novelty of their current work, the authors write in the introduction of the present manuscript that extending the existing measurement approach to molecules is not straightforward. After reading the manuscript, which is very clear and sound from the experimental point of view, I am not sure I spotted where the authors achieved a considerable advance over the previous result. The measurement technique used in this work is identical to the previously published one. It could be that the discussion and interpretation of the measured data, which I found to be a bit confusing, also precluded me from discerning the novelty of this work. Based on the current state of the manuscript, I suggest its publication in a more specialized journal, e.g. Communication Physics, after the revision.

Below you will find my questions mainly related to figure 4 of the manuscript. Analyzing their experimental data, the authors claim that applying their STM lifetime measurement technique to ZnPc and H2Pc yields two surprising outcomes: In particular, the effective barrier height of NaCl (linear slopes in Fig3) seem to be the same for both the PIR and NIR species on all surfaces. Secondly, the PIR and NIR have substantially different lifetimes (vertical offset in Fig3b) on Au(111). The authors then rationalize these observations qualitatively using a double barrier picture that includes the valence and conductance bands of NaCl and the upshifted Shockley states of Au(111) and Cu(111) surfaces.

On Cu(111), the authors claim that the neutralization of both charge states proceeds via the LUMO resonance because if it were not the case, the effective barrier height would have been different for two different charge states. This situation demands that HOMO and LUMO resonances are inside the bias window applied to induce the PIR (see Fig 4b). While this type of level alignment could be possible, I would expect that the resonances enter the bias window sequentially upon increasing the bias, which should give an experimental possibility of proving the claimed level arrangement. Do experimental data support the proposed interpretation?

On Au(111), it looks more confusing. First, Fig 4c suggests that the relaxation of the NIR state involves both HOMO and LUMO, so I get the same question: Is there an experimental indication for the case of both resonances being inside of the applied bias window during the measurement? The text also does not clearly state the alignment of which level (HOMO or LUMO) defines the effective barrier height for NIR on Au(111). I feel the authors assert that the LUMO position with respect to the CBM determines the barrier, but I am not sure.

Also, as far as I comprehend, the fact that the effective barrier height for PIR is similar to the one of the NIR on Au(111) is explained by the hole tunnelling through the HOMO aligned close to the VBM of NaCl (Fig.4d). Here I get a question: Cu(111) case in Fig4b and Au(111) case in Fig.4d seem to look identical concerning the HOMO to the VBM alignment. Why wasn't the "hole tunnelling" argument also not used for Cu(111)?

Reviewer #3 (Remarks to the Author):

Review of "Charge-state lifetimes of single molecules on ultrathin insulating films" by Kaiser et al. In this study the authors have measured lifetimes of charge-state of a single luminescent molecules adsorbed on NaCl films grown on Cu and Au substrates. The experimental findings are explained based on the simple model of tunneling and the energy-level alignment at the interfaces.

The measurements are of top quality, and the paper is written clearly. The technique used here has been developed in a proceeding work of the same author (ref. 31 of the MS), however, the focus on this paper lies in experimental determination of the charge-state lifetime of the important molecular system especially in STM-induced electroluminescence.

I have a reservation on broad impact that warrants publication in Nat. Commun. Since the paper is written primarily to give important information to a specific field, STM and STM-luminescence, I am concerned that it may not have an impact on researchers in other fields. Mainly out of this concern, I am not positive for publication of the manuscript, at least in this form, in Nat. Commun. Some detailed discussions and criticisms are provided below for improving the manuscript.

1. In a previous study using Cl vacancies as samples (ref. 31 of the MS), saturation currents were 1-2 orders of magnitude larger than in the present experiment of adsorbed molecules. I believe that it would be important to discuss this difference for generalizing their results and at the same time the authors are responsible for that. The authors stated in the introduction "Cl-vacancy states are localized within the top-most NaCl layer and possess an s-wave character as well as strong lateral confinement". Since molecules have a significantly larger area than CL defects, one would expect the conductivity between the molecule and the substrate to be larger, but the results were quite the opposite.

2. The authors found similar barrier heights for the neutralization of anion and cation on both substrates. They explained this with the relative energetic position of CBM and VBM of the NaCl film with respect to the tunneling channels. However, they did not show any evidence or support for this. How were the position of CBM and VBM determined? Otherwise, they cannot draw the summarizing figures (Fig. 4).

3. The authors stated "Interestingly, the spatial distribution of regions of increased current does not resemble the HOMO density but rather the LUMO density (see Fig. 5c for comparison)." However, it is not easy especially for non-experts to find the resemblance. They should provide more clear description. I found that the number of bright lobes is different for HOMO and the newly observed image (lower right panel in Fig. 5d).

4. dI/dV spectra of molecules showing PIR and NIR should be provided in Fig. 1 or SI. It would be helpful for people to better understand the experiment.

5. Regarding the previous comment and comment #2, dI/dV spectra of NaCl showing CBM and VBM should be provided in SI, if possible.

6. In P 7 L7-8 from the bottom, Fig. 4a and 4b might be 5a and 5b.

Reviewer #4 (Remarks to the Author):

The authors demonstrate an interesting strategy to measure the lifetimes of intermediate molecular charged states that are related to the tunneling process from/to either the tip or substrate in an STM junction. Specifically, the response of the transient charged states is reflected by the saturation behavior in the steady-state current. By examining the dependence of charged state lifetimes on the NaCl thickness and the type of substrate, they reveal that the tunneling process also pertains to the energy level alignment in the whole molecular junction. In addition, the authors also propose that additional tunneling channels can become available due to higher-order charging process. Based on the novelty presented in this work, the manuscript can be

considered for publishing in Nature Communications if the authors can properly address the following minor comments or concerns.

1. To better understand the energy level alignment, dI/dV spectra are desirable. Perhaps, they should be provided in the main text so that the energies of the PIR and NIR states can be easily identified.

2. As shown by previous works (such as Doppagne, et al. Science 361 (2018) 251-255 and Chen, et al. PRL 122 (2019) 177401), the charging and discharging process can form excitons and trions, which are excited neutral and charged states. Can the authors explain why the influence of these excited states can be omitted in the dynamics when deducing the expression for the saturation current?

3. The unit of the current in figure 2a and b can be changed to pA to be consistent with that in figure 2d.

Typos:

1. The time constants illustrated in Fig. 1a are supposed to be τ_c and τ_d rather than T_c and T_d .

2. In page 5 line 10, the increasing factors when adding a monolayer NaCl for Cu(111) and Au(111) should be 18 and 30 respectively according to $\exp(m)$.

3. In page 7 line 20 and 21, Fig. 4 should be Fig. 5.

Reviewer #5 (Remarks to the Author):

In their manuscript, Kaiser et al. present an experimental STM study revealing the lifetimes of charge states of prototypical organic molecules on thin insulating NaCl layers adsorbed on either Cu(111) or Au(111) surfaces. By performing current-versus-distance spectroscopy, that is, by measuring the tunneling current, I , as a function of tip height, z , the authors determine the discharging lifetime for tunneling from the molecule through the NaCl layer into the metal substrate from a plateau in the $I(z)$ curves. While such an approach has already previously been applied for Cl-vacancy states in NaCl layers, the extension to molecular states has not been reported before and provides novel insights into the mechanisms of charge dynamics at a single molecule level which is of great significance for the field. The applied experimental methodology is of high standard, and the data analysis and interpretation of the results is sound. Therefore, I recommend publication in Nature Communications after the authors have considered the minor points listed below.

(1) The red lines for the fits in Fig. 2a and b are hardly visible.

(2) The energy-level diagram depicted in Figure 4 is used to rationalize the experimental findings, particularly the differences between Cu(111) and Au(111) and the different lifetimes of the cation and anion on Au(111). However, it is not quite clear, if this diagram is merely of qualitative nature or to what extent the depicted energy positions reflect a quantitative picture. Can the authors explain in more detail which experimental data are used that allowed them to arrive at the energy level diagrams of Figure 4a-d? Are there any computational data that support their arguments?

(3) Page 7: In the paragraph starting with "The dication ...", the reference should be to Figure 5 not Figure 4.

Reply to the reviewers

We thank the reviewers for their detailed comments on our paper. Below, you will find our response to the questions and comments raised by the reviewers, with the changes implemented in the manuscript and the SI highlighted in grey.

The revised version of the manuscript as well as the supporting information includes the corresponding changes, highlighted in grey and partly by 'Track Changes' and comments.

Reviewer #1 (Remarks to the Author):

The work "Charge-state lifetime of single molecules on ultrathin insulating films" investigates transport through phthalocyanine molecules deposited on 3-5ML of NaCl on Au(111) and Cu(111). It brings compelling evidence that the lifetimes of charged molecular states excited through the resonant transport channels of the system depend fundamentally on the conductivity across the NaCl spacer and consequently govern formation of excited states in the chromophore. The authors put the results into perspective with previous results on Cl-vacancies and evaluate the tunnelling barriers and tunnelling rates (lifetimes) quantitatively.

This is an interesting study that has the potential to shed some light on the recently discussed topics in the STML community. It is a surprisingly solid result considering the simple method employed for this study. However, the discussion is very dense and certain points seem quite confusing. I would recommend a thorough revision of the interpretation and possibly adding more data, before entering the second round. Here are my comments and questions that may need to be reflected in the manuscript.

We thank the reviewer for recognizing the importance of our work and for the comments. Below, we address comments and questions and highlight the corresponding changes that were made in the manuscript and supporting information.

1) Ballistic transport of electrons is not considered in the analysis. It may be obvious to the authors, but which are the grounds to disregard it?

Ballistic transport usually refers to the conduction of charge carriers without scattering. In the context of tunneling through a double barrier junction, we are not entirely sure to which aspect exactly reviewer 1 refers to as ballistic transport of electrons. If this question is related to the contribution of electrons tunneling directly from the tip to the metal (without being attached nor losing energy to the molecule), we neglect this tunnel channel because of our experimental observations and data. The distinct plateau in the $I(z)$ spectra shows that the contribution of direct tunneling between tip and metal substrate, which will exponentially increase with decreasing z , is negligible. This can be rationalized by the large tunneling distance – through the added vacuum and NaCl barriers – for the direct tunneling path between tip and metal substrate.

We edited the MS, page 4 first paragraph: "For small z , $I(z)$ shows a plateau at the saturation current I_{sa} . This shows that at the electronic resonances tunneling between tip and metal substrate is governed by a two-step tunnel process via the molecule, whereas the contribution of direct tunneling, which should increase exponentially with decreasing z , is negligible. In addition, in the regime of saturation the tunnel current is governed by tunneling through the NaCl barrier, and the molecule is charged most of the time."

2) Band models in Fig.1a and in Fig.4 raise these concerns:

i) As drawn in these schemes, voltage drop seems to be constant, regardless of the bias and vacuum barrier width. This is commented in the part 2 of the SI, and stated that upon reducing the vacuum barrier, the change is expected to be small. At the same time, the total barrier height is *assumed* to be 10-20% at NaCl, citing DOI:10.1038/s41565-018-0087-1 (which puts the value at 17% for 14 MLof NaCl, which seems rather inconsistent with the current manuscript). The problem here is that there is no convincing data for this quite a striking assumption of a constant voltage drop.

In particular, positions of NIR and PIR are not documented by any STS measurements in this work. It would be a good idea to add them to the SI. It can (dis)prove the independence on the z-distance and layer thickness, as it has been done in DOI:10.1103/PhysRevB.98.201403.

Evaluating the ratio of the exact voltage drop across NaCl and vacuum is challenging. It has been estimated in DOI:10.1038/s41565-018-0087-1 (Ref. 15 of MS) and shown there, that even the exact tip geometry enters (see Fig. S6 in DOI:10.1038/s41565-018-0087-1). Here, with the 10% to 20% of the applied voltage, we roughly estimated the range of voltage drops for the different geometries used in this current work. We neither assume that it is constant as a function of bias (as it is a relative voltage drop) nor that it is constant as a function of distance (the relative voltage drop across NaCl will decrease as the tip height increases, for a specific NaCl thickness). The voltage drop would be critical for the analysis if it could shift a molecular resonance outside the bias voltage window (e.g., by varying z). However, the voltages in the experiment were chosen such that this can be excluded. Beyond this precaution, estimating the relative voltage drop exactly is not the scope of our work. Importantly, the exact amount of the voltage drop ratio across vacuum and NaCl does not affect the analysis and statements made in the manuscript. To make that clear we added the following statements:

In the manuscript, page 4 last paragraph: “Bias voltage dependent measurements reveal that τ_d remains the same within a few 100 mV around the peak voltage of the resonance and several 100 mV above it (see Fig. S3a, b). By changing the total applied bias voltage (and tip height), the voltage drop across NaCl also changes by a small amount [34]. I_{sat} being independent of V (for voltages associated to a certain ion resonance, see Fig. S3a, b) and of the tip height z (Fig. 2a, b) indicates that the charge-state lifetime does not vary significantly as a function of the voltage drop across NaCl for a specific ion resonance. A more detailed discussion of the voltage dependence and the partial voltage drops in vacuum and NaCl is given in the Supporting Information S2.”

In the SI in S2: “The voltage drop across the tip-sample junction is composed of the voltage drop over the vacuum gap (between tip and molecule) and the one over the NaCl film (between molecule and metal substrate). The ratio of these two voltage drops depends on the exact geometry, i.e., the tip-molecule distance z, the tip position, the thickness of the NaCl film d_{NaCl} , and the tip-radius. Following the estimation given in ref. [1], we estimate the relative voltage drop over NaCl to be in the range of 10-20% of the total applied bias voltage for the geometries present in this study.

Upon applying a bias, the voltages of the molecular ion resonances are shifted with respect to the zero bias condition by the voltage drop across the NaCl film. As a consequence, changes in z or d_{NaCl} directly affect the voltages of the ion resonances. To avoid shifting of the molecular ion resonances outside the voltage window when changing the lever arm (z or d_{NaCl}), we typically used absolute bias voltages slightly larger (up to a few 100 mV larger) than the corresponding peak positions in dI/dV , V_{PIR} and V_{NIR} . One example for the shift due to the changed lever arm is shown in Fig. S2d, where we show $dI/dV(V)$ -curves recorded at top H_2Pc adsorbed on 3 and 4 MLof NaCl on Au(111). The peaks and onsets of PIR and NIR shift

to higher absolute values on 4 ML NaCl, as a result of the larger relative voltage drop across NaCl on 4 ML compared to 3 ML

A similar trend is expected for decreasing z (i.e., a decrease in z will lead to an increase in voltage drop across NaCl and thus an up-shift of the absolute values of V_{PIR} and V_{NIR}) [2]. Unaffected by the changed lever arm, the saturation current, i.e., the charge-state lifetime, does not vary with decreasing z (see, e.g., Fig. 2b in the main text). In addition, we recorded bias dependent $I(z)$ curves (Fig. S3), showing that, also independent of the applied bias voltage, the charge-state lifetime remains the same. This indicates that the charge-state lifetime, and with it the barrier height for the corresponding tunneling process through NaCl, does not vary significantly as a function of the voltage drop across NaCl.”

We also changed Fig. 1, not to make the false impression of assuming a constant voltage drop across NaCl and included Figure S2d in the SI.

Figure S1: $dI/dV(V)$ recorded on different systems. (a) H_2Pc on 3 ML NaCl / Au(111) (setpoint $I = 0.5$ pA, $V = 2$ V, $\Delta z = -1$ Å). (b) H_2Pc on 4 ML NaCl / Cu(111) (setpoint $I = 0.5$ pA, $V = 1.6$ V, $\Delta z = -1$ Å). (c) ZnPc on 3 ML NaCl / Au(111). (d) Comparison of H_2Pc on 3 ML and on 4 ML NaCl / Au(111) (setpoint $I = 0.5$ pA, $V = 2$ V, $\Delta z = -1$ Å). The onsets of PIR and NIR are indicated by the dashed lines.

Concerning the comparison with the cited publication (15 in MS): The voltage drop over NaCl is estimated to be on the order of 10-20% of the total voltage V_{applied} , based on a simple plate capacitor model with estimated values for the tip-molecule distances using the formula given in DOI:10.1038/s41565-018-0087-1 (Equation (6) of SI). In addition, as also shown in the previous publication, the plate capacitor geometry yields an overestimation of the relative voltage drop. Considering a spherical geometry of the tip apex, as it is done in DOI:10.1038/s41565-018-0087-1, results in smaller values of the relative voltage drop as estimated with the plate-capacitor model. The reason that here for 3 to 5 ML NaCl, we estimate similar values for the voltage drop as compared to said publication, comes from the relatively large tip-molecule distance used in DOI:10.1038/s41565-018-0087-1 (2 nm vacuum gap in the reference compared to about 0.5 - 1.0 nm here). This will, roughly, compensate the thicker NaCl-layer that was used in the cited publication: The vacuum gap is about a factor of 2-4 larger, and the NaCl thicknesses are a factor of 3-5

larger in DOI:10.1038/s41565-018-0087-1, compared to the different geometries used in the work under review.

ii) Upon charge transfer between an electrode and an orbital, and following exciton, energy level renormalization should occur, but it is not considered and not discussed at all. Instead the shift of the PIR and NIR is interpreted solely as a result of the voltage drop. This contradicts some previous works where the role of the intramolecular Coulomb interaction is suggested as the main reason e.g. in DOI:10.1021/acs.nanolett.8b04484

As the referee mentions correctly, there are several mechanisms that influence or shift the level alignment. First of all, one has to discriminate between the energetic positions of orbitals of the neutral case and those, in which the orbitals are (temporarily) occupied and the molecule becomes charged. To make this distinction, in the latter case the energy levels are often called transport levels, implying that Coulomb interactions are accounted for. Irrespectively, the voltage drop will change the level alignments of tip, substrate and molecule as discussed above. Finally, after exciton formation or charging, the levels will shift further because of screening. In the case of charging, this shift is attributed to the reorganization energy.

In the context of the above, it is important to note that the sketched level alignments in Fig. 4 schematically depict the energy levels after the charging event. That is in Fig. 4a,c the molecule is charged negatively and in Fig. 4b,d the molecule is charged positively (e.g., after a charge transfer between tip and molecule). The shown level alignment already includes the reorganization energies, hence it depicts the situation with screening. Because of the questions and comments of the referees, we now included the lifted degeneracy of the spins in anions and cations, thus allowing us to energetically distinguish transitions to S_1 and T_1 and to provide estimated quantitative schematics in Fig. 4 instead of the qualitative ones in the previous version.

The charging will affect the energetic positions of the former HOMO, LUMO, SOMO and SUMO levels of anion and cation, and their energetic positions are challenging to access experimentally. Note that for naphthalocyanine such levels of a (positively) charged molecule have been measured on thick insulating films, indicating reorganization energies on the order of several 100 meV (DOI: 10.1038/s41565-018-0087-1). Here, we schematically draw the levels, based on our spectroscopy but also based on several literature values and observations from STML experiments, indicating possible pathways for neutralization of the charged systems.

To make this clear we discussed Figure 4 and the paragraphs related to charge transfer from the sample in more detail:

In the main text, p. 6 l. 15ff: “In the following, we will discuss these observations based on the energetic alignment of potential tunnel channels that may be involved in the neutralization process with respect to the substrate states (Fig. 4). Note that the picture presented in Fig. 4 is a sketch with estimations based on previous experiments (see Fig. S5), which could be refined and better quantified by future experiments and theoretical investigations.

In general, neutralization of a charged species in a $D_0^{+/-}$ state can, depending on the level alignment between molecule and metal substrate, proceed via several channels, i.e., in addition to transitions to the neutral ground state (S_0) transitions to excited states (e.g., T_1 , S_1) could be energetically allowed [3,4,9,38]. Because only a small fraction of the applied bias drops in the NaCl film, about 10-20% for the used geometries (see, e.g., [15] and Supporting Information S2), by changing the applied bias voltage we can

shift the molecular levels only within a very limited energy window with respect to the substrate's electrochemical potential. Instead, we can use metal substrates with different work functions to compare situations where only the channel to the neutral ground state is energetically allowed, to situations where also channels to excited states are possible. Although the exact positions of the anion's/cation's electronic levels are not accessible in STM on ultrathin insulating films, we can deduce for which cases excited states can be formed in the neutralization step, e.g., from comparison to STML measurements and comparing the energies of the PIR and NIR with excited state energies (i.e., energies of optical transitions). This information allow us to draw an estimated picture of the level alignments between the transiently charged molecules and the substrate. We assume that on Cu(111) the anion, formed at the NIR at about $V_{\text{NIR}} = 1$ V, only neutralizes to the ground state S_0 , whereas the cation, formed at the PIR at about $V_{\text{PIR}} = -2.2$ V, can also form the excited S_1 and T_1 state upon neutralization. On the other side, we assume that on Au(111) the cation, formed at the PIR at about $V_{\text{PIR}} = -1.1$ V, only neutralizes to S_0 , whereas the anion, formed at the NIR at about $V_{\text{NIR}} = 2.1$ V, can also form the excited S_1 and T_1 state [5,38]. For a more detailed discussion see the Supporting Information S3 and Fig. S5. This energetic alignment of the different channels is considered and reflected in the schematics shown in Fig. 4, and it allows us to discuss how the lifetime changes when different channels are open for neutralization. Note that the level alignments in Fig. 4 depict the molecular resonances after the charging event via the tip, which differ from those of the neutral molecule (Fig. 2e, f) due to, e.g., Coulomb repulsion, lifted spin degeneracy and reorganization energy [7,15].

The similar m for anion and cation indicates similar effective barrier heights for the charged-to-neutral transitions. If here the effective barrier height simply corresponded to the respective energetic difference to the vacuum level (or another single fixed energy level), the extracted barrier height Φ_{NaCl} would be significantly different for the PIR (neutralization of the cation) compared to the NIR (neutralization of the anion). Note that in general, for the NIR (PIR) neutralization occurs by electrons tunnelling at energies above (below) the chemical potential of the metal sample, i.e., to unoccupied (from occupied) sample states. The similar m can be rationalized by the effective NaCl barrier height Φ_{NaCl} resulting from the relative energetic position of the tunnel channels with respect to both the conduction band minimum (CBM) and the valence band maximum (VBM) of the NaCl film. For tunnel processes involving states that are energetically near the VBM, hole tunneling, for which the tunnel barrier height is given with respect to the VBM and thus decreases with decreasing carrier energy, can dominate [39]. Specifically, for the cation (Fig. 4b, d), the small energetic difference of one of the tunnel channels to the VBM could explain its similar m compared to the anion, which is energetically closer to the CBM and vacuum level (Fig. 4a, c). A quantitative description of tunneling through NaCl beyond the above considerations would require the complex-valued band structure of NaCl including the band-gap region [40,41] as well as the different potential landscape because of the different transient charge state of the molecule in the initial state of the tunnelling process.”

In the SI, p. 6: “S3. Decharging from the substrate

Figure 4 in the main text schematically depicts, in a single-electron picture, the molecular energy levels of ZnPc/H₂Pc after charging from the tip. Note that the single-electron picture is used here to depict the level alignment of the channels with respect to, e.g., VB, CB and IS. Although the energy levels derive from the neutral molecule's HOMO and LUMO, their energetic positions are different from the neutral molecule's levels because of, e.g., Coulomb interaction, the lifted spin degeneracy, and reorganization [1,3]. The exact energetic positions of these levels, corresponding to transiently charged states, cannot be probed in our experiment. However, we can estimate the energetic position of the cation's (anion's) singly unoccupied (occupied) frontier orbital with respect to the tip's electrochemical potential, based on the positions of the PIR (NIR) peak in dI/dV and estimating the reorganization energy. From these levels

(SOMO of anion at NIR and SUMO of cation at PIR), we construct the levels corresponding to the transitions to S_1 and T_1 states by using reported energies of luminescence and phosphorescence, respectively. The positions of the lowest unoccupied level at NIR and the highest occupied level at PIR, are estimated from experiments, in that molecules were doubly negatively charged on thick NaCl films [4, 14], with consideration of the lever arm [3]. The charging energy E_{charge2} for doubly charged states, corresponding to the additional energy needed from the single charging event to double charging, is estimated here as $E_{\text{charge2}} = 1.2$ eV for both dianions and dications.

In addition, to corroborate the accessibility of possible channels to excited states in the neutralization step, we compared our conclusions to previous STML experiments on these systems. If a molecule shows luminescence in STML at PIR or NIR (on a given surface) the transition from the corresponding charged D_0 state back to the neutral charge state can entail the transition to S_1 [5–8]. Hence, if at a given bias voltage luminescence can be observed for the system, the channel to the S_1 state is accessible for the neutralization of the molecule.

Based on these arguments, we sketch in Fig. 4 the energetic positions of the molecular single-electron energy levels of the charged molecule with respect to tip and sample states, taking into account several literature values and observations from STML experiments, indicating possible pathways for the neutralization of the singly charged systems.”

Figure S5 depicts the quantities that were taken into account for the level alignment presented in Fig. 4 in the main text, using the examples of the anion at bias voltages that correspond to the NIR (Fig. S5a) and the cation at voltages corresponding to the PIR (Fig. S5b) on Au(111). The sample voltages for these conditions, i.e., at NIR and at PIR, correspond to the respective peaks in dI/dV . As a result of reorganization, the anion’s (cation’s) energy levels are shifted down (up) by the reorganization energy E_{reorg} with respect to the corresponding energy levels of the neutral molecule. We assume here a reorganization energy of $E_{\text{reorg}} = 0.4$ eV. The energy levels are broadened due to electron-phonon coupling [2,9]; the total peak width, being the energy range within which tunneling into the molecular orbitals is appreciable in the experiment, is here assumed to be $w = 0.6$ eV. The energetic difference

Figure S5: Sketch depicting the estimated quantitative alignment of the anionic (a) and cationic (b) energy levels with respect to sample and tip states. The assumed values for reorganization energy are $E_{\text{reorg}} = 0.4$ eV, charging energy for doubly charged states $E_{\text{charge2}} = 1.2$ eV, and linewidth of the energy levels $w = 0.6$ eV, indicated by the grey shaded areas around the molecular levels. The energetic difference between the molecular levels are associated with the transition energies of the S_0-S_1 and S_0-T_1 transitions, respectively. The onset of the IS for NaCl/Au(111) is $V = -0.27$ V. All shown energy differences are to scale.

between the molecular levels are associated with the transition energies of the S_0 - S_1 and S_0 - T_1 transitions, respectively.

The energies for the S_1 S_0 transition in H_2Pc and $ZnPc$ are 1.81 eV and 1.89 eV, respectively [5,10]. T_1 energies of 1.24 eV for H_2Pc and 1.14 eV for $ZnPc$ have been reported for molecules in chloronaphthalene solution [11,12]. Note, however, that the presence of the substrate and the tip is known to shift the energies of optical transitions with respect to the values in solution [13,14], and phosphorescence from T_1 of H_2Pc or $ZnPc$ in STM has not been reported as of now.

Note that for the PIR on Au(111), the applied bias voltages are close to being sufficient to facilitate the formation of the T_1 upon neutralization from the substrate. However, the exact energies of the T_1 S_0 transition for H_2Pc and $ZnPc$ adsorbed on an ultrathin insulating film atop a metal substrate are not known. We tentatively excluded the tunnel channel for T_1 formation in Fig. 4d based on the energetic arguments given above. However, it cannot be fully ruled out because of the uncertainty of the energy of the T_1 S_0 transition and the significant level broadening on NaCl.”

In addition, we added Figure S5 to explain the level alignment.

Note also that reviewer 2 had similar comments and questions. Hence, for changes implemented in the manuscript comments from both reviewers were taken into account.

iii) The role of the interface state is probably not needed to explain the lower saturation currents for Au(111) PIR, because the tunnelling barrier alone for the corresponding electron transfer is the highest among all the considered cases.

The tunnel barrier for the neutralization of the molecule affects the exponent (slope) of the $I(d_{NaCl})$ ($\ln(I(d_{NaCl}))$) curves. We find almost identical slopes for both resonances (NIR and PIR) on Au but still different lifetimes (see Fig. 3b). Thus, the significant differences in the saturation currents between PIR (longer lifetimes) and NIR (shorter lifetimes) cannot be explained by differences in the barrier height. However, the participation of the interface state (in the NIR process, but not in PIR) explains the vertical shift of the graphs in Fig. 3b and the significantly longer lifetimes of the process that involves the PIR compared to the NIR on Au(111).

To make this important finding of our paper clearer we added a sentence in the paragraph discussing the different charge-state lifetimes of anion and cation on Au(111), p. 8 first paragraph: “Since we find similar slopes in $\ln(I_{sat})$ for the discharging of anion and cation from the sample, the second observation, i.e., the cation lifetimes on Au(111) being about one order of magnitude longer than that of the anions, cannot be explained by differences in the barrier height for the involved tunneling events. Instead, the observed differences can be rationalized by considering the sample’s local density of states (LDOS) at the energy of the ion resonances.”

Remarks to consider:

Shouldn't the slope equation be correctly: $m = \ln(1/I_{sat})/d_{NaCl}$?

The reviewer is correct in that the sign of the slope does not fit with our convention of the equation. With our definition of the equation, i.e., $m = \ln(I_{sat})/d_{NaCl}$, the slope takes a negative value, which would be consistent with the way we plot $I_{sat}(d_{NaCl})$ in Fig. 3.

Action taken: The values for the slopes are corrected accordingly, i.e., now showing negative values.

While the title, abstract and introduction mention charge-state lifetimes, the τ_d is named as "discharging time" near to the main equation. It could be better to name it consistently throughout the paper.

Action taken: τ_d was named consistently "charge-state lifetime" throughout the paper.

It will be very interesting to put the charge-state lifetime into context with the past STML measurements that measured "lifetimes", e.g. DOI:10.1038/s41467-017-00681-7, DOI:10.1021/acsnano.1c01318.

Action taken: A comparison with previously measured lifetimes was included in the manuscript, p. 10 last paragraph:

"Previously, Hanbury Brown-Twiss interferometry and phase fluorometry in combination with STML have been used to access exciton dynamics in single molecules adsorbed on ultrathin insulating films [27,29]. The extracted lifetimes comprise time constants for both excitation and decay of molecular excitons and are of similar magnitude as the charge-state lifetimes reported here (on the order of 500 ps for 3MLNaCl). This indicates that, for certain geometries, the neutralization process from the substrate is the rate limiting process in the formation and decay of molecular excitons and, thus, the previously extracted time constants are likely dominated by the charge-state lifetime."

Fig.S3 depicts the scheme of possible states and transitions between them. In the light of recent discussions in the community, the position of the states at the 'Free energy' axis could be set quite precisely according to the onsets of the PIR and NIR energies. Also, in the many-body picture, direct transition $S_0 \rightarrow D_1^+$ is possible upon charge transfer between the tip and the molecule.

We agree with the reviewer that indeed it is possible to indicate the position of the PIR and NIR states on the 'Free Energy' axis.

Concerning the accessibility of D_1^+ from S_0 , the reviewer is correct in that, from an energy conservation perspective, this transition is possible at an applied bias voltage of -2.5 V on Au(111) and -3.8 V on Cu(111). We did not discuss the direct transition from S_0 to D_1^+ because the data recorded at these elevated bias voltages show an increase in current above I_{sat} only at small z (large tunnel rates between tip and molecule). This indicates that the process leading to this increase in current is of higher order, so for example a two-electron process.

In order to correct and clarify this point, we modified Fig. S4 (was Fig. S3 in the first version of the manuscript) and added the direct S_0 to D_1^+ transition. In addition, we removed the electron configurations of the states that are indicated in the figure since especially for D_1^+ this is not known and can lead to potentially wrong assumptions about the nature of transitions between the states.

We also added a short paragraph explaining why for the explanation of the current increase at high bias voltage, one-electron transitions are not taken into account, see p. 4 in the SI: "Note that here we do not discuss transitions to states that become directly accessible from S_0 in a single-electron tunneling event (faded arrows in Fig. S4) for the following reason: The increase in current above I_{sat} and change in contrast in c.h. STM only appears at small z , i.e., sets in only at a large tunnel rate between tip and molecule. This indicates that the underlying process is of higher order (i.e., a two- or more-electron process) and involves tunneling into a transient state of the molecule (e.g., T_1 or D_0^+) and hence only becomes accessible if charge transfer from the tip happens at comparable or faster time scales than the depopulation of the involved transient state."

Figure S4: Energy diagram of ZnPc on Au(111), including many-body transitions and including higher-lying states and possible many-body transitions that become accessible at increased absolute voltage values. Blue arrows indicate charge-state transitions by tunneling between molecule and sample (through the NaCl barrier), and red arrows by tunneling between molecule and tip (through the vacuum barrier). Single-electron transitions from the neutral ground state S_0 (faded red lines) are not considered for the observed increase in current at small z . Yellow arrows indicate transitions that do not involve a change in charge state, such as radiative transitions. The dashed red arrows indicate charge-state transitions by tunneling through the vacuum barrier, in which electrons are removed from higher-lying orbitals (the LUMO of the neutral molecule). These transitions could lead to the observed contrast in constant-height STM images at $V = -2.5$ V and small tip height. Note that this diagram is simplified and does not contain additional, even higher-lying excited states of the system.

Reviewer #2 (Remarks to the Author):

The manuscript "Charge-state lifetimes of single molecules on ultrathin insulating films" presents the lifetime measurement of the positive- and negative-ion resonance states of ZnPc and H2Pc molecules adsorbed on Au(111) and Cu(111) and rationalizes the dependence of the measured lifetimes on the thickness of the NaCl decoupling layer. As the authors mention, the experimental STM technique used for the measurements is not new and was already applied by one of the co-authors to determine the lifetimes of Cl vacancies on NaCl overlayers of varying thicknesses. Trying to defend the novelty of their current work, the authors write in the introduction of the present manuscript that extending the existing measurement approach to molecules is not straightforward. After reading the manuscript, which is very clear and sound from the experimental point of view, I am not sure I spotted where the authors achieved a considerable advance over the previous result. The measurement technique used in this work is identical to the previously published one. It could be that the discussion and interpretation of the measured data, which I found to be a bit confusing, also precluded me from discerning the novelty of this work. Based on the current state of the manuscript, I suggest its publication in a more specialized journal, e.g. Communication Physics, after the revision.

The reviewer is correct that the used technique is not new. What is new is that it has been applied to molecules, also that it has been applied to positive ion resonances in addition to negative ion resonances. And it is new that it has been applied to metals with different work functions. The variation in this parameter-space lets us draw conclusions about the different effects on the charge-state lifetimes, such as the involvement of the valence band onset and the interface state, going significantly beyond the previous work, and providing reference values for the charge-state lifetimes of multiple relevant molecular systems.

Below you will find my questions mainly related to figure 4 of the manuscript. Analyzing their experimental data, the authors claim that applying their STM lifetime measurement technique to ZnPc and H2Pc yields two surprising outcomes: In particular, the effective barrier height of NaCl (linear slopes in Fig3) seem to be the same for both the PIR and NIR species on all surfaces. Secondly, the PIR and NIR have substantially different lifetimes (vertical offset in Fig3b) on Au(111). The authors then rationalize these observations qualitatively using a double barrier picture that includes the valence and conductance bands of NaCl and the upshifted Shockley states of Au(111) and Cu(111) surfaces.

On Cu(111), the authors claim that the neutralization of both charge states proceeds via the LUMO resonance because if it were not the case, the effective barrier height would have been different for two different charge states. This situation demands that HOMO and LUMO resonances are inside the bias window applied to induce the PIR (see Fig 4b). While this type of level alignment could be possible, I would expect that the resonances enter the bias window sequentially upon increasing the bias, which should give an experimental possibility of proving the claimed level arrangement. Do experimental data support the proposed interpretation?

Note also that reviewer 1 had in parts similar comments and questions. Hence, for changes implemented in the manuscript both were taken into account. For changes related to Fig. 4 in the main text we therefore also refer reviewer 2 to the changes highlighted in the answers to 2.ii from reviewer 1.

It is important to note that Fig. 4 shows the levels for the charged molecule, that is, after charging by tunneling between tip and molecule. (See also comment to reviewer 1, question 2). The levels are derived from the HOMO and LUMO levels of the neutral molecule, but the respective levels of the anion and cation will shift in energy with respect to the neutral molecule, predominantly due to intra-molecule electron-

electron Coulomb repulsion. Because of that, the HOMO-derived and the LUMO-derived levels are both inside the bias voltage window for the anionic molecule as displayed in Fig. 4b, but for the neutral molecule only the HOMO level would be inside the bias voltage window. Hence, only because of the first tunneling process to the tip, the LUMO becomes available for neutralization in the second step. The threshold for the entire process is therefore set by the voltage, at which the first tunneling process becomes available.

It is also important to note that most of the applied voltage, about 80% to 90%, will drop across the vacuum barrier and only 10 to 20% across NaCl (which is reflected in the observed same plateaus obtained for different voltages, see Fig. S2 and our response to the 1st question of reviewer 1). By changing the applied voltage one can sweep the Fermi level of the tip with respect to the molecule's electronic levels, but one can only shift in a very small window the molecular levels against the metal substrate's Fermi level. That is, the energy levels of the molecule for the neutralization cannot be sequentially accessed by sweeping them against the Fermi level of the metal substrate, as proposed by the referee. On Cu(111) if the voltage is large enough to detach an electron from the HOMO in the first step (leading to the situation sketched in Fig. 4b), the LUMO of the cation will always be accessible by tunneling from the substrate, because of the level alignment on Cu(111). This is different on Au(111), where, due to the larger work function of Au(111) compared to Cu(111), the LUMO in the cation cannot be accessed from the substrate (at voltages near the PIR). To summarize: We wanted to do what the reviewer proposed, that is, seeing the sequential onset of the channels for neutralization via tunneling between substrate and molecule. However, we cannot simply sweep that voltage drop across the NaCl surface by changing the applied bias to sequentially access the different channels for neutralization for a given system. Thus, to "sweep" the bias across the NaCl film we varied the work function of the metal. Comparing Au(111) to Cu(111), we observe the effect of the additional channel via the LUMO for Cu(111).

As last sentence of the introduction we explained this motivation for the different work-function surfaces:

"Comparing results on the different work function surfaces Au(111) and Cu(111) allows us to shift the molecular electronic states with respect to the sample's electrochemical potential in order to probe the effect of different possible tunnel channels for the neutralization event."

On Au(111), it looks more confusing. First, Fig 4c suggests that the relaxation of the NIR state involves both HOMO and LUMO, so I get the same question: Is there an experimental indication for the case of both resonances being inside of the applied bias window during the measurement?

Yes, there are experimental indications that both channels are possible at the NIR on Au(111). The answer to this question goes in line with the previous one. In Fig. 4c, the channels for neutralization of the charged molecule are sketched. In STM on ultrathin insulating films, we cannot sweep the levels of the molecule with respect to those of the substrate to see the onset of different channels for neutralization. However, from STML we can deduce which resonances are involved in the neutralization: At the NIR on Au, exciton formation in ZnPc and H₂Pc has been shown to be possible (see, e.g., Doležal et al., Nano Lett. 19 (2019) or Rai et al., Nano Lett. 20 (2020)), while at PIR, only the transition back to the neutral ground state S₀ is energetically accessible. Thus, at the NIR on Au(111) there are at least two tunnel channels accessible (to the ground state and to the excited state).

We added paragraphs to clarify the level alignment to the main text and the SI, see answer to reviewer 1, question 2.ii.

The text also does not clearly state the alignment of which level (HOMO or LUMO) defines the effective barrier height for NIR on Au(111). I feel the authors assert that the LUMO position with respect to the CBM determines the barrier, but I am not sure.

In general, if several channels for neutralization are possible, the fastest ones should determine the saturation current and the charge-state lifetime. To that end, we also expect to measure the barrier height for that (fastest) process. At first glance, for the situation sketched in Fig. 4c one would expect the process with the highest energy (into the LUMO) to have the lowest barrier and being the fastest, however, since the tunneling occurs through NaCl where the quantitative tunneling rates are governed by the detailed band structure, it is not obvious which of the processes is fastest. The fact that exciton formation in STML on this system has been observed via the NIR [e.g., Doležal et al., Nano Lett. 19 (2019)], suggests that tunneling from the molecule's HOMO to the substrate (in Fig. 4c) constitutes a non-negligible contribution to the overall tunnel current. However, for this system we cannot exclude that more than one channel contributes significantly (have decay times of similar order of magnitude). Note that for the situation displayed in Fig. 4c, in all accessible channels, tunneling via the interface state is possible, such that these considerations do not touch the provided explanation based on the interface state.

Changes: We modified the text describing Figure 4 to improve clarity and mention explicitly that for the situation in Fig. 4c we do not know which of the two paths is dominant.

In the main text, p. 8 last paragraph: “In summary, on both surfaces, the neutralization of the anion and cation by tunnelling through the NaCl film, exhibits very similar barrier heights, although the energetic separation to the vacuum level of the involved tunnel channels is very different. This leads us to the conclusion that Φ_{NaCl} results from the relative energetic position of the tunnel channels with respect to both the CBM and the VBM of the NaCl film. In addition, on Au(111), the longer lifetimes observed at the PIR compared to the NIR – despite similar barrier heights – indicate tunnelling across different distances. This can be rationalized by the increased LDOS due to the IS, that contributes to at least one channel in all situations (Fig. 4a,b,c) except for the PIR on Au(111) (Fig. 4d).

As a side remark, if several channels are open, i.e., as for the PIR on Cu(111), see Fig. 4b, and for the NIR on Au(111), see Fig. 4c, the fastest channels should dominantly govern the charge-state lifetime and the measured barrier height. However, in our experiment we cannot separately measure the rates of competing channels. Based on our results and arguments discussed above we would expect that the channels that involve the IS, sketched bold in Fig. 4, are faster and thus dominant. STML showing light emission from the S_1 state for the NIR on Au(111) [5,17,38], indicates that out of the three channels that involve the IS, shown in Fig. 4c, the channel that is lowest in energy contributes at least significantly.”

Also, as far as I comprehend, the fact that the effective barrier height for PIR is similar to the one of the NIR on Au(111) is explained by the hole tunnelling through the HOMO aligned close to the VBM of NaCl (Fig.4d). Here I get a question: Cu(111) case in Fig4b and Au(111) case in Fig.4d seem to look identical concerning the HOMO to the VBM alignment. Why wasn't the "hole tunnelling" argument also not used for Cu(111)?

We considered that argument, and as the referee suggests would expect a similar barrier height for the channel via the HOMO of the PIR on Cu(111) (lowest transition in Fig. 4b) compared the only channel of the transition on Au(111), (transition in Fig. 4d). However, on Cu(111) also another transition is possible which involves the IS (highest transition on Fig. 4b). As we argue, the transition involving the IS leads to shorter lifetimes (as we find, compared to the case of Au) and thus dominates over the slower transition on Cu(111).

Action taken: We included a more thorough discussion of the involved states, see answer to previous point.

Reviewer #3 (Remarks to the Author):

Review of “Charge-state lifetimes of single molecules on ultrathin insulating films” by Kaiser et al. In this study the authors have measured lifetimes of charge-state of a single luminescent molecules adsorbed on NaCl films grown on Cu and Au substrates. The experimental findings are explained based on the simple model of tunneling and the energy-level alignment at the interfaces.

The measurements are of top quality, and the paper is written clearly. The technique used here has been developed in a proceeding work of the same author (ref. 31 of the MS), however, the focus on this paper lies in experimental determination of the charge-state lifetime of the important molecular system especially in STM-induced electroluminescence.

I have a reservation on broad impact that warrants publication in Nat. Commun. Since the paper is written primarily to give important information to a specific field, STM and STM-luminescence, I am concerned that it may not have an impact on researchers in other fields. Mainly out of this concern, I am not positive for publication of the manuscript, at least in this form, in Nat. Commun. Some detailed discussions and criticisms are provided below for improving the manuscript.

We thank the reviewer for recognizing the quality of our measurements and for the comments. In the following, we address the comments and questions in detail.

1. In a previous study using Cl vacancies as samples (ref. 31 of the MS), saturation currents were 1-2 orders of magnitude larger than in the present experiment of adsorbed molecules. I believe that it would be important to discuss this difference for generalizing their results and at the same time the authors are responsible for that. The authors stated in the introduction “Cl-vacancy states are localized within the top-most NaCl layer and possess an s-wave character as well as strong lateral confinement”. Since molecules have a significantly larger area than Cl defects, one would expect the conductivity between the molecule and the substrate to be larger, but the results were quite the opposite.

Comparing the results on single molecules with those obtained on Cl-vacancies, one needs to consider two things: (1) Cl-vacancies are located within the NaCl-film whereas molecules are adsorbed on top with an adsorption height of approximately 0.3 nm [Miwa et al., Phys. Rev. B 93 (2016)]. Hence, the tunnel rate for discharging of a molecule adsorbed on top of N layers of NaCl, compared to that of a Cl-vacancy in layer N, should be strongly reduced because of a larger distance to the underlying metal. (2) As the reviewer pointed out correctly, Cl-vacancy states are strongly localized compared to molecular orbitals. Related to that, resonances of a Cl-vacancy lie at higher energies. As a consequence, the tunnel barrier height is reduced compared to the case of adsorbed molecules, and thus, the tunnel probability should be generally higher. Both effects point towards a lower tunnel probability between molecule and substrate as compared to Cl-vacancy and substrate, which fits our experimental findings. With respect to the argument that a larger lateral overlap region of wave functions should provide a larger tunneling matrix element, one should keep in mind that the confined wave function needs to be normalized to one. Hence, a more confined wave function has larger (local) absolute values.

We added a short paragraph, bringing our findings into context with the charge-state lifetimes reported for Cl-vacancies, p. 5 second paragraph: “The charge-state lifetimes τ_a range from around 50 ps (3ML NaCl/Cu(111)) to 20 ns (5ML NaCl/Au(111)). The previously reported charge-state lifetimes of the anionic charge states of Cl-vacancies in NaCl [31] are about one order of magnitude smaller (on the order of 1 ns for 5ML NaCl) compared to the charge-state lifetimes of the investigated negatively charged molecules on 5 ML NaCl. Aside from other effects, e.g. different barrier heights and energetic positions of the

resonances, this relates to the probed Cl-vacancies being located within the top monolayer of the NaCl-film, whereas molecules are adsorbed on top of the NaCl film with an adsorption height of approximately 0.3 nm [35].”

2. The authors found similar barrier heights for the neutralization of anion and cation on both substrates. They explained this with the relative energetic position of CBM and VBM of the NaCl film with respect to the tunneling channels. However, they did not show any evidence or support for this. How were the position of CBM and VBM determined? Otherwise, they cannot draw the summarizing figures (Fig. 4).

We support this claim by the barrier height that we extracted from the $I_{\text{sat}}(d_{\text{NaCl}})$ curves. Since the barrier heights for all tunnel processes yield approximately the same results, independent of the transient charge of the molecule, we conclude that the barrier height for tunneling between molecule and substrate is not given by the vacuum level or the conduction band onset alone. Instead, to explain the similar slope m for the case of the PIR on Au(111) (Fig. 4d), also the onset of the valence band of the NaCl needs to be considered. As these are qualitative arguments, we do not need to know the exact positions of the energetic levels.

We modified manuscript and SI in the following way:

We changed a sentence in the figure caption of Fig. 4: “The shown level alignment qualitatively corresponds to literature values of the samples’ work functions and the measured voltages of the ion resonances deduced from the positions of the neutral molecule’s ion resonances and STML. The dashed lines indicate which single electron states derive from the HOMO (lower pair of states) and LUMO (higher pair of states) of the neutral molecule, the corresponding α and β spin channels are indicated.”

We added a paragraph on pages 6 to 8 to improve the explanation of Fig. 4, and we added a chapter to the SI to explain the assessment of the level alignment in Fig. 4 in more detail (see also the answer to point 2.ii of reviewer 1, and the answer to the second-last point of reviewer 2).

3. The authors stated “Interestingly, the spatial distribution of regions of increased current does not resemble the HOMO density but rather the LUMO density (see Fig. 5c for comparison).” However, it is not easy especially for non-experts to find the resemblance. They should provide more clear description. I found that the number of bright lobes is different for HOMO and the newly observed image (lower right panel in Fig. 5d).

We added a more detailed description of the constant current image presented in Fig. 5, p. 9: “Interestingly, the spatial distribution of regions of increased current shows a ring of 12 maxima, as on a clock, with bright maxima at positions 3, 6, 9, 12 o’clock and two equally spaced fainter maxima in between. This corresponds to the shape and symmetry observed for the LUMO rather than the HOMO density, which exhibits three (not two) equally spaced fainter lobes between the bright lobes (see Fig. 5c for comparison).”

4. dI/dV spectra of molecules showing PIR and NIR should be provided in Fig. 1 or SI. It would be helpful for people to better understand the experiment.

Action taken: We added dI/dV spectra in Fig. 2 and the SI, see also answer to referee #1, point 2.i.

5. Regarding the previous comment and comment #2, dI/dV spectra of NaCl showing CBM and VBM should be provided in SI, if possible.

Action taken: we added dI/dV spectra of NaCl on Au(111) and Cu(111) to the SI in Fig. S6.

Figure S6: $dI/dV(V)$ on 3 ML NaCl / Cu(111) (black curve, setpoint $V = -0.2$ V, $I = 2$ pA, $\Delta z = 2$ Å) and 4 ML NaCl / Au(111) (grey curve, setpoint $V = -5$ V, $I = 2$ pA, $\Delta z = 0$ Å). The curves were obtained by numerical differentiation of $I(V)$ curves.

6. In P 7 L7-8 from the bottom, Fig. 4a and 4b might be 5a and 5b.

We corrected the wrong reference.

Reviewer #4 (Remarks to the Author):

The authors demonstrate an interesting strategy to measure the lifetimes of intermediate molecular charged states that are related to the tunneling process from/to either the tip or substrate in an STM junction. Specifically, the response of the transient charged states is reflected by the saturation behavior in the steady-state current. By examining the dependence of charged state lifetimes on the NaCl thickness and the type of substrate, they reveal that the tunneling process also pertains to the energy level alignment in the whole molecular junction. In addition, the authors also propose that additional tunneling channels can become available due to higher-order charging process. Based on the novelty presented in this work, the manuscript can be considered for publishing in Nature Communications if the authors can properly address the following minor comments or concerns.

We thank the reviewer for recognizing the novelty of the work and recommending it for publication in Nature Communications, as well as the comments. In the following, we address the comments and questions in detail.

1. To better understand the energy level alignment, dI/dV spectra are desirable. Perhaps, they should be provided in the main text so that the energies of the PIR and NIR states can be easily identified.

Action taken: We added dI/dV spectra in Fig. 2 and the SI.

2. As shown by previous works (such as Doppagne, et al. Science 361 (2018) 251-255 and Chen, et al. PRL 122 (2019) 177401), the charging and discharging process can form excitons and trions, which are excited neutral and charged states. Can the authors explain why the influence of these excited states can be omitted in the dynamics when deducing the expression for the saturation current?

The referee touches an interesting point. We argue that if the radiative lifetime of an excited state is shorter than the rate limiting tunneling process, one can safely neglect excited states. In this scenario, the saturation current is given by the net charge transfer between tip and sample, and whether or not an excited state is formed does not affect this charge-transfer rate. (Nonetheless, tunneling pathways leading to excited states must still be considered, as we do here). We expect luminescence from singlet excited states to be faster than the typical tunneling rates observed here.

A triplet excited state may indeed have a radiative lifetime longer than the involved tunneling rates. However, we argue that even this would not lead to significant changes. An excited state can still decay via tunneling pathways. An excited state has – per definition – a higher energy than the ground state. Therefore, more tunneling pathways are available as compared to the respective ground state. Hence, we do not expect any current blocking situations occurring from excited states. Instead, any long-lived excited state will quickly decay via a fast tunneling channel, leaving behind the system in the ground state to proceed with the cycle of tip-molecule and molecule-substrate tunneling.

Exciton formation needs to be considered to understand the different charge-state lifetimes, in the discussion of the barrier heights and Fig. 4. Here, we consider that neutralization of the molecule can lead to the formation of an exciton (possible in Fig. 4b and c) and that the corresponding tunnel channel can show a different tunnel rate compared to the one corresponding to the transition directly back into the neutral ground state.

The reviewer is also correct in pointing out that trions can be formed. Trion formation was neglected for the main discussion and in Fig. 4, since the bias voltages that were used are not sufficiently high to lead to efficient trion formation [see e.g., Doppagne et al., Science 361 (2018); Hung et al., arXiv:2210.11118 (2022)]. This is different for the data in Fig. 5, where the formation of the trion (D_1^+) is discussed and can explain the decreased charge-state lifetime observed at elevated voltages.

3. The unit of the current in figure 2a and b can be changed to pA to be consistent with that in figure 2d.

As Fig. 1a, b, shows the current in a log scale we prefer to show this in units of A. To read off 10^4 pA is probably more confusing than 10^{-8} A. For the rest of the MS, where no log-scales are used we stick to pA.

Typos:

1. The time constants illustrated in Fig. 1a are supposed to be τ_c and τ_d rather than T_c and T_d .
2. In page 5 line 10, the increasing factors when adding a monolayer NaCl for Cu(111) and Au(111) should be 18 and 30 respectively according to $\exp(m)$.
3. In page 7 line 20 and 21, Fig. 4 should be Fig. 5.

We corrected the typos.

Reviewer #5 (Remarks to the Author):

In their manuscript, Kaiser et al. present an experimental STM study revealing the lifetimes of charge states of prototypical organic molecules on thin insulating NaCl layers adsorbed on either Cu(111) or Au(111) surfaces. By performing current-versus-distance spectroscopy, that is, by measuring the tunneling current, I , as a function of tip height, z , the authors determine the discharging lifetime for tunneling from the molecule through the NaCl layer into the metal substrate from a plateau in the $I(z)$ curves. While such an approach has already previously been applied for Cl-vacancy states in NaCl layers, the extension to molecular states has not been reported before and provides novel insights into the mechanisms of charge dynamics at a single molecule level which is of great significance for the field. The applied experimental methodology is of high standard, and the data analysis and interpretation of the results is sound. Therefore, I recommend publication in Nature Communications after the authors have considered the minor points listed below.

We thank the reviewer for recognizing the novelty and significance of our work, and for the comments. A detailed discussion of the questions and comments is provided below.

(1) The red lines for the fits in Fig. 2a and b are hardly visible.

We modified Figure 2 accordingly.

(2) The energy-level diagram depicted in Figure 4 is used to rationalize the experimental findings, particularly the differences between Cu(111) and Au(111) and the different lifetimes of the cation and anion on Au(111). However, it is not quite clear, if this diagram is merely of qualitative nature or to what extent the depicted energy positions reflect a quantitative picture. Can the authors explain in more detail which experimental data are used that allowed them to arrive at the energy level diagrams of Figure 4a-d? Are there any computational data that support their arguments?

The diagram in Fig. 4 shows schematics of the energy level alignment of the charged molecules. These levels derive from the neutral molecule's HOMO and LUMO, but their energetic positions are different (e.g., because of Coulomb repulsion, relaxation, and lifted spin degeneracy). In STM on ultrathin insulating films, the energy levels of the transiently charged molecule are not directly accessible. However, the alignment can be estimated from combining available experimental data from our work and the literature, namely of the resonances with respect to the sample's/tip's electrochemical potential, the known optical gaps, measurements of reorganization energy, and by findings from STML. As an example: If a molecule shows luminescence in STML at PIR or NIR (on a given surface), the transition from the populated charged state back to the neutral ground state can proceed via the neutral excited state S_1 . This means that, in this case, at least two tunnel channels are accessible for the neutralization of the molecule. For more details, we refer to our replies to reviewers 1 and 2.

We clarified the paragraphs describing the improved Figure 4 and added a paragraph to the SI explaining the level alignment of Fig. 4, see also answers to point 2.ii of reviewer 1 and the second-last point of reviewer 2.

The following changes made to the SI (SI, p. 6) are also listed in the reply to question 2.ii of reviewer 1: "Figure 4 in the main text schematically depicts, in a single-electron picture, the molecular energy levels of ZnPc/H₂Pc after charging from the tip. Note that the single-electron picture is used here to depict the level alignment of the channels with respect to, e.g., VB, CB and IS. Although the energy levels derive from the neutral molecule's HOMO and LUMO, their energetic positions are different from the neutral molecule's levels because of, e.g., Coulomb interaction, the lifted spin degeneracy, and

reorganization [1,3]. The exact energetic positions of these levels, corresponding to transiently charged states, cannot be probed in our experiment. However, we can estimate the energetic position of the cation's (anion's) singly unoccupied (occupied) frontier orbital with respect to the tip's electrochemical potential, based on the positions of the PIR (NIR) peak in dI/dV and estimating the reorganization energy. From these levels (SOMO of anion at NIR and SUMO of cation at PIR), we construct the levels corresponding to the transitions to S_1 and T_1 states by using reported energies of luminescence and phosphorescence, respectively. The positions of the lowest unoccupied level at NIR and the highest occupied level at PIR, are estimated from experiments, in that molecules were doubly negatively charged on thick NaCl films [4, 14], with consideration of the lever arm [3]. The charging energy $E_{\text{charge}2}$ for doubly charged states, corresponding to the additional energy needed from the single charging event to double charging, is estimated here as $E_{\text{charge}2} = 1.2$ eV for both dianions and dications.

In addition, to corroborate the accessibility of possible channels to excited states in the neutralization step, we compared our conclusions to previous STIML experiments on these systems. If a molecule shows luminescence in STIML at PIR or NIR (on a given surface) the transition from the corresponding charged D_0 state back to the neutral charge state can entail the transition to S_1 [5–8]. Hence, if at a given bias voltage luminescence can be observed for the system, the channel to the S_1 state is accessible for the neutralization of the molecule.

Based on these arguments, we sketch in Fig. 4 the energetic positions of the molecular single-electron energy levels of the charged molecule with respect to tip and sample states, taking into account several literature values and observations from STIML experiments, indicating possible pathways for the neutralization of the singly charged systems.

Figure S5 depicts the quantities that were taken into account for the level alignment presented in Fig. 4 in the main text, using the examples of the anion at bias voltages that correspond to the NIR (Fig. S5a) and the cation at voltages corresponding to the PIR (Fig. S5b) on Au(111). The sample voltages for these conditions, i.e., at NIR and at PIR, correspond to the respective peaks in dI/dV . As a result of reorganization, the anion's (cation's) energy levels are shifted down (up) by the reorganization energy E_{reorg} with respect to the corresponding energy levels of the neutral molecule. We assume here a reorganization energy of $E_{\text{reorg}} = 0.4$ eV. The energy levels are broadened due to electron-phonon coupling [2,9]; the total peak width, being the energy range within which tunneling into the molecular orbitals is appreciable in the experiment, is here assumed to be $w = 0.6$ eV. The energetic difference between the molecular levels are associated with the transition energies of the S_0 - S_1 and S_0 - T_1 transitions, respectively.

The energies for the S_1 - S_0 transition in H_2Pc and $ZnPc$ are 1.81 eV and 1.89 eV, respectively [5,10]. T_1 energies of 1.24 eV for H_2Pc and 1.14 eV for $ZnPc$ have been reported for molecules in chloronaphthalene solution [11,12]. Note, however, that the presence of the substrate and the tip is known to shift the energies of optical transitions with respect to the values in solution [13,14], and phosphorescence from T_1 of H_2Pc or $ZnPc$ in STIML has not been reported as of now.

Note that for the PIR on Au(111), the applied bias voltages are close to being sufficient to facilitate the formation of the T_1 upon neutralization from the substrate. However, the exact energies of the T_1 - S_0 transition for H_2Pc and $ZnPc$ adsorbed on an ultrathin insulating film atop a metal substrate are not known. We tentatively excluded the tunnel channel for T_1 formation in Fig. 4d based on the energetic arguments given above. However, it cannot be fully ruled out because of the uncertainty of the energy of the T_1 - S_0 transition and the significant level broadening on NaCl.

(3) Page 7: In the paragraph starting with “The dication ...”, the reference should be to Figure 5 not Figure 4.

We corrected the reference in the paragraph.

Reviewer comments, second round

Reviewer #1 (Remarks to the Author):

The manuscript "Charge-state lifetimes of single molecules on ultrathin insulating films" has been updated following the concerns noted in the review report. Changes have been introduced addressing most of the raised issues. Questions have been satisfactorily discussed. Therefore I support the publication of the manuscript in the present form.

Reviewer #3 (Remarks to the Author):

2nd Review of "Charge-state lifetimes of single molecules on ultrathin insulating films" by Kaiser et al.

I appreciate that the authors considered my comments. However, still I cannot recommend publication of this manuscript mainly because their explanation regarding the similar slopes in Fig. 3 is not well supported. I strongly recommend the authors to reconsider the following points.

1. Regarding my previous comment #2, the authors answered "We support this claim by the barrier height that we extracted from the $I_{\text{sat}}(d_{\text{NaCl}})$ curves.". This is obviously logically strange because the observed barrier height is a part of things that the author is trying to explain. They observed similar slopes in Fig. 2, and hypothesized that the observation could be explained by the similar barrier heights between the resonance tunneling channel and either CBM or VBM. Then, I asked for evidence for the positions of CBM and VBM. However, the authors didn't show any evidence.

The author also answered "As these are qualitative arguments, we do not need to know the exact positions of the energetic levels.". I agree that we don't need exact positions of energetic levels. However, at least the values of energetic positions of CBM and VBM must be supported by evidence (experimental or theoretical), even if not exact. Otherwise, Figure 4 is entirely the author's imagination and is not a scientific argument.

2. The authors added Fig. S6 (dI/dV spectra of NaCl) to suggest band edges (CBM and VBM). However, I believe the argument here is not scientifically sound. For example, it is well-known that on the positive voltage side of NaCl films on metal substrates strong image potential state appear around 3 V and above, which is not the CBM. I strongly ask the authors to refine the data and argument here. Showing CBM and VBM should be very important support to the author's explanation.

Reviewer #4 (Remarks to the Author):

The authors have made proper responses to the comments raised by the referees and revised the manuscript accordingly. I therefore recommend its publication in Nature Communications.

Reviewer #5 (Remarks to the Author):

The authors have responded convincingly to all questions and comments raised in the first round and revised the manuscript and SI accordingly. Therefore, I can recommend the publication of the manuscript in its current form in Nature Communications.

Response letter

We thank reviewers #1, 4, and 5 for their recommendation for publication in Nature Communications, and reviewer #3 for their detailed comments on our paper. Below, you will find our response to the questions and comments raised by reviewer #3, with the changes implemented in the manuscript and the SI highlighted in grey.

The revised version of the manuscript as well as the supporting information includes the corresponding changes, highlighted in grey and partly by 'Track Changes' and comments.

In addition to the reviewer's comments, we have modified Figure 4 in the manuscript to remove duplicate labeling of the interface state (IS), and added a missing 'z' to the equation given in Section S4 of the Supporting Information.

Reviewer #3 (Remarks to the Author):

I appreciate that the authors considered my comments. However, still I cannot recommend publication of this manuscript mainly because their explanation regarding the similar slopes in Fig. 3 is not well supported. I strongly recommend the authors to reconsider the following points.

1. Regarding my previous comment #2, the authors answered "We support this claim by the barrier height that we extracted from the $I_{sat}(d_{NaCl})$ curves." This is obviously logically strange because the observed barrier height is a part of things that the author is trying to explain. They observed similar slopes in Fig. 2, and hypothesized that the observation could be explained by the similar barrier heights between the resonance tunneling channel and either CBM or VBM. Then, I asked for evidence for the positions of CBM and VBM. However, the authors didn't show any evidence. The author also answered "As these are qualitative arguments, we do not need to know the exact positions of the energetic levels." I agree that we don't need exact positions of energetic levels. However, at least the values of energetic positions of CBM and VBM must be supported by evidence (experimental or theoretical), even if not exact. Otherwise, Figure 4 is entirely the author's imagination and is not a scientific argument.

2. The authors added Fig. S6 (dI/dV spectra of NaCl) to suggest band edges (CBM and VBM). However, I believe the argument here is not scientifically sound. For example, it is well-known that on the positive voltage side of NaCl films on metal substrates strong image potential state appear around 3 V and above, which is not the CBM. I strongly ask the authors to refine the data and argument here. Showing CBM and VBM should be very important support to the author's explanation.

We thank the referee for their candid criticism and understand that we have to make our argumentation and claims even more clear in the paper. We regret that our previous answer could be misunderstood as a circular reasoning.

It is our experimental result, that the slope m of $\ln(I_{sat}(d_{NaCl}))$ is similar for both resonances (PIR and NIR). This slope corresponds to the effective barrier height. Therefore, we experimentally show similar effective barrier heights for these processes. In the MS we made sure to always have inserted "effective", when we refer to the experimentally determined effective barrier height.

In the description of the exponential decay in tunneling, three levels of sophistication can be distinguished. In the strongest simplification, one would define a barrier potential, and the barrier height is the difference between this potential and the electron's energy. Such a barrier potential could be assumed to coincide with the CBM. In our experiment, the electrons' energies for the

different considered tunneling processes differ by more than one electronvolt, yet they experimentally exhibit very similar effective barrier heights, as pointed out above. This discrepancy clearly demonstrates that this picture is far too simple. Note that, the latter observation is completely independent from the exact position (or shape) of the barrier potential.

The second level of sophistication would take into consideration that within the valence band, electrons are described by propagating wave functions, such that within the valence band the tunneling barrier should completely vanish. Thus, the effective barrier height depends on the position of the electron's energy relative to both, the CBM and the VBM. Since the barrier height must vanish at both, the CBM and the VBM, it must exhibit a maximum somewhere in the band gap. At such an extremum it barely varies with energy – in line with our experimental observations.

At the third level of sophistication, one would describe electrons inside the band gap with evanescent wave functions, hence, with a complex-valued wave vector. This full quantum description of the tunneling problem reveals that tunneling does not necessarily fall off with the decay constant corresponding to the simple picture provided by the one-dimensional Schrödinger equation with a given barrier potential. This is not surprising, because inside a solid, the potential strongly varies at shortest length scales in all three dimensions and electron-electron interaction complicates matters further. Such a description is established in other fields of research and is referred to as complex band structure (see *e.g.* Phys. Rev. B 98, 195422). The qualitative arguments outlined above for the second level of sophistication remain valid, and in such complex-band-structure one can directly see that deep in the band gap, the imaginary component of the wave vector becomes typically quite independent of energy, see *e.g.* Phys. Rev. B 98, 195422, in very good agreement with our observations.

While unfortunately, there seems no complex-band-structure calculation for the specific case of NaCl available in the literature, we allude to all the above in our manuscript. In Fig. 4 we would like to augment the text description of the above by a schematic picture, illustrating the need to consider the NaCl's band structure. This we do by showing not only a single barrier potential, as it is often done, but by representing the NaCl with indicating the CB and VB.

To summarize, the barrier potential and thus the effective barrier height observed in STM for tunneling events through the NaCl depends on its complex-valued band structure. While this includes the conduction and valence band, the effective barrier height is not directly given by their respective onset energies. Important for the qualitative understanding of the experimental observation of an effective barrier height being nearly independent of energy is mainly, that all tunneling processes occur at energies deep inside the wide band gap.

There is indeed previous experimental evidence of the latter. The combination of PRB 17, 2537 (1978) and the PhD thesis of Stefan Fölsch, University of Hannover (1991) indicate that the CBM is located close to the vacuum level and that even for films of only few atomic layers the band gap of NaCl is (almost) fully developed. Hence, the band gap should extend from roughly 4-5 eV below the Fermi level to 3.5-4.5 eV above it.

We agree with the referee that the observation of a strong increase of the current in STM for thin films around 3.5 V is not enough to claim that the CBM is located there, but we have additional data on thick films of NaCl that also indicate a conductance onset of the film at around 4 V, in agreement with the above literature values (see for example Fig. 1 in Steurer *et al.*, Appl. Phys. Lett. 104 (2014)).

To conclude this point, we reiterate that for vacuum tunneling the barrier height can well be approximated by the energy difference between tunnel channel and vacuum level. Our results show

that this is very different for tunneling through NaCl. We are convinced that this is an important experimental result and has not been described so far, to the best of our knowledge. The statement that this is caused by alignment with respect to not only the vacuum level, but to the entire band structure is our hypothesis for this effect.

To make our point clear we clarified the discussion in this regard and explicate what is our speculation and what are our experimental results (highlighted changes):

“The experimentally observed similar m for anion and cation indicates similar effective barrier heights for the charged-to-neutral transitions. If here the effective barrier height simply corresponded to the respective energetic difference to the vacuum level (or another single fixed energy level, e.g., the conduction band minimum or valence band maximum), the extracted barrier height Φ_{NaCl} would be significantly different for the PIR (neutralization of the cation) compared to the NIR (neutralization of the anion). Note that in general, for the NIR (PIR) neutralization occurs by electrons tunnelling at energies above (below) the chemical potential of the metal sample, i.e., to unoccupied (from occupied) sample states. We hypothesize that the similar m result from the effective NaCl barrier height Φ_{NaCl} being influenced by the relative energetic position of the tunnel channels with respect to both the conduction band minimum (CBM) and the valence band maximum (VBM) of the NaCl film. It has been shown that for tunnel processes involving states that are energetically near the VBM, hole tunneling, for which the tunnel barrier height is given with respect to the VBM and thus decreases with decreasing carrier energy, can dominate [39]. Specifically, for the cation (Fig. 4b, d), the small energetic difference of one of the tunnel channels to the VBM could explain its similar m compared to the anion, which is energetically closer to the CBM and vacuum level (Fig. 4a, c). However, since the different neutralization processes involve tunnel channels with different energetic alignment with respect to the substrate’s states and thus with respect to CBM and VBM (see Fig. 4), it seems unlikely that the effective barrier heights result from the energetic separation of these channels to either VBM or CBM alone, but rather that both bands can contribute to the effective barrier height. Important for the qualitative understanding of the experimental observation of a nearly energy independent effective barrier height seems to be that all contributing tunneling processes occur at energies deep inside the wide band gap of NaCl. A quantitative description of tunneling through NaCl beyond the above considerations would require the complex-valued band structure of NaCl including the band-gap region [40–42] as well as the different potential landscape because of the different transient charge state of the molecule in the initial state of the tunnelling process.”

In addition, we modified the caption of Fig. 4 to reinforce that the sketched CBM and VBM only serve as an illustration to highlight the role of the NaCl’s band structure for tunneling, and furthermore providing estimated values for VBM and CBM (further detailed in section S5), as the referee requested:

“Figure 4: Transition from a molecule’s transiently charged state to the neutral charge state by tunneling through the NaCl barrier. The frontier molecular levels, chemical potentials of tip (μ_t) and sample (μ_s), and the interface state (IS) are indicated. The shown level alignment corresponds to voltages of the ion resonances deduced from the positions of the neutral molecule’s ion resonances and STML. The dashed lines indicate which single electron states derive from the HOMO (lower pair of states) and LUMO (higher pair of states) of the neutral molecule, the corresponding α and β spin channels are indicated. The grey shaded area indicates the linewidth of the levels. Because of different work functions, the vacuum-level aligned molecular ion resonances are shifted to larger bias values on Au(111) compared to Cu(111) [12,32]. Thicker (thinner) arrows indicate channels that involve (do not involve) the IS. In a)-c), the IS contributes to at least one channel, while in d), it does

not. A more detailed depiction of how the shown level alignment was derived is shown in the SI in Fig. S5. By schematically showing the conduction band minimum (CBM) and valence band maximum (VBM) of NaCl we indicate the need to consider the NaCl's band structure for tunneling. References [42] and [66] suggest that the CBM is roughly aligned with the vacuum level while the band gap is at least 8 eV (see section S5 in SI).”

Concerning the determination of the band onsets from Fig. S6 in the SI, we agree with the referee that our measurements can only serve as a lower bound. We mention that image potential states (added references [15,16]) might contribute.

As we detailed above, our arguments do not require knowledge of the position of VBM and CBM. However, in response to the referee's request of providing these values, we estimated them from literature and added this information to the improved and expanded final section of the SI: “S5. Estimation of valence band maximum and conduction band minimum” on p. 8 in the SI:

S5. Estimation of valence band maximum and conduction band minimum

“Figure S6 shows $dI/dV(V)$ spectra recorded atop the bare NaCl surface for different underlying metal substrates, ~~showing the onset of tunneling~~. Note that the measured apparent onsets of tunnelling depend on the tip-NaCl distance and might relate to image potential states [15,16] and only serve as a rough lower bound for the band onsets ~~in the situation present in the double-barrier tunnel junction with a molecule below the tip where this distance is larger~~. The combination of ref. [17] and ref. [18] indicate that the CBM is located close to the vacuum level and that even for films of only few atomic layers the band gap of NaCl is (almost) fully developed, *i.e.*, about 8.5 eV. The NaCl film lowers the metal work function, *i.e.*, 4.9 eV for Cu(111) and 5.3 eV for Au(111), by about 1 eV [19,20]. We estimate that the band gaps, that is VBM to CBM, should extend from roughly 4 - 5 eV below the Fermi level to roughly 3.5 - 4.5 eV above it.”

Related to the expanded discussion of the CBM and VBM and the complex band structure of NaCl, we added the following references to the main text:

[42] F. J. Himpsel and W. Steinmann, *Angle-Resolved Photoemission from the NaCl (100) Face*, Phys. Rev. B 17, 2537 (1978).

[66] Fölsch, Stefan, *Elektronenspektroskopische Untersuchungen Zur H₂O-Adsorption Auf Der NaCl(100) Oberfläche*, University Hannover, 1991.

And to the SI:

[15] H.-C. Ploigt, C. Brun, M. Pivetta, F. Patthey, and W.-D. Schneider, *Local Work Function Changes Determined by Field Emission Resonances: Na Cl/ Ag (100)*, Phys. Rev. B 76, 195404 (2007).

[16] Q. Guo, Z. Qin, C. Liu, K. Zang, Y. Yu, and G. Cao, *Bias Dependence of Apparent Layer Thickness and Moiré Pattern on NaCl/Cu(001)*, Surface Science 604, 1820 (2010).

[17] F. J. Himpsel and W. Steinmann, *Angle-Resolved Photoemission from the NaCl (100) Face*, Phys. Rev. B 17, 2537 (1978).

[18] Fölsch, Stefan, *Elektronenspektroskopische Untersuchungen Zur H₂O-Adsorption Auf Der NaCl(100) Oberfläche*, University Hannover, 1991.

[19] R. Bennewitz, M. Bammerlin, M. Guggisberg, C. Loppacher, A. Baratoff, E. Meyer, and H.-J. Güntherodt, *Aspects of Dynamic Force Microscopy on NaCl/Cu(111): Resolution, Tip-Sample Interactions and Cantilever Oscillation Characteristics*, Surf. Interface Anal. **27**, 462 (1999).

[20] C. Loppacher, U. Zerweck, and L. M. Eng, *Kelvin Probe Force Microscopy of Alkali Chloride Thin Films on Au(111)*, Nanotechnology **15**, S9 (2004).

We respectfully reject the comment “Otherwise, Figure 4 is entirely the author's imagination and is not a scientific argument”. We detailed, in the first revised version, how Fig. 4 is quantitatively derived (see Fig. S5 and section S3 in the SI) from experimental results and established literature. We acknowledge that there are estimations, *e.g.*, on the level broadening, which we state in the corresponding text in the SI. But we are confident that these are well supported and sound scientific arguments, which led us to the schematic pictures drawn in Fig. 4. And actually, we are convinced that this picture will be adopted and will foster understanding of the corresponding effects in the community. We hope that with the above discussion we can settle the point about the VBM and CBM.

We hope that our answer convinces referee #3 to recommend our paper for publication.

Sincerely,

Katharina Kaiser

on behalf of all authors

Reviewer comments, third round

Reviewer #3 (Remarks to the Author):

Comments with a figure are given by the PDF file attached.

3rd Review of “Charge-state lifetimes of single molecules on ultrathin insulating films” by Kaiser et al.

I appreciate that the authors considered my comments. Now I understand how the authors drew Fig. 4.

The vacuum level (or work function) is lowered by roughly 1 eV by the formation of the NaCl film.

CBM locates roughly 1 eV below the vacuum level (from the PhD thesis of Stefan Fölsch), so roughly 2 eV below the vacuum level of the metal substrates.

Assuming the band gap of 8.5 eV, the VBM position is determined.

The authors' logic is reasonable.

Request 1: Add in SI a detailed explanation for how the Fig. 4 was drawn so that anyone can draw the corresponding figures in other systems in the same way.

I point out an important inaccuracy in the figure.

The band gap between CBM and VBM in Fig. 4 is not scaled.

Based on my measurement, the distance between the lines of CBM and VBM are corresponding only 7.3 eV much smaller than 8.5 eV.

Request 2: Redraw the figure with correctly scaled band gap of NaCl (see the roughly corrected figure below).

With this scaled figure, I found it's difficult to believe the effective barrier heights of the NaCl barrier in Au(111) at NIR and at PIR are similar. The apparent energy barrier is very different.

The apparent barrier in the PIR case is ~ 4.1 eV, which is almost twice larger than that in the NIR case (~ 2.3 eV).

From my perspective, if the authors insist the effective barrier heights for the PIR and NIR on Au(111) are similar, it is necessary to explain or discuss why the effective barrier height becomes similar despite the very different apparent barrier height.

The authors stated in the revised manuscript "However, since the different neutralization processes involve tunnel channels with different energetic alignment with respect to the substrate's states and thus with respect to CBM and VBM (see Fig. 4), it seems unlikely that the effective barrier heights result from the energetic separation of these channels to either VBM or CBM alone, but rather that both bands can contribute to the effective barrier height. Important for the qualitative understanding of the experimental observation of a nearly energy independent effective barrier height seems to be that all contributing tunneling processes occur at energies deep inside the wide band gap of NaCl." However, this does not explain the above thing nor justify not to explain it, in my opinion.

Request 3: Explain or at least discuss why the effective barrier height becomes similar despite the very different apparent barrier heights of PIR and NIR in Au(111) case.

In summary, the challenging experiment was conducted with the highest level of technique and setup. The results are new, surprising and impressive, which is clearly different from the vacuum tunneling barrier. Their hypothesis is simple and reasonable but is not well supported. I am not convinced that the results are reasonably explained. Therefore, I do not recommend the publication of the current form of the manuscript in Nature communications.

Response letter

We thank the reviewer for the fast turnaround and their effort and comments. While we have implemented some of the requested changes, we believe that there are still some fundamental misunderstandings, that we would like to further clarify, including in the manuscript and the SI. In the following, we would like to provide a more detailed response to the reviewer's requests and comments.

3rd Review of "Charge-state lifetimes of single molecules on ultrathin insulating films" by Kaiser et al.

I appreciate that the authors considered my comments. Now I understand how the authors drew Fig. 4. The vacuum level (or work function) is lowered by roughly 1 eV by the formation of the NaCl film. CBM locates roughly 1 eV below the vacuum level (from the PhD thesis of Stefan Fölsch), so roughly 2 eV below the vacuum level of the metal substrates. Assuming the band gap of 8.5 eV, the VBM position is determined. The authors' logic is reasonable.

We thank the reviewer for their effort to reproduce our reasoning regarding the energetic position of the CBM. However, we realized that apparently the previous Fig. 4 led to the wrong impression that we assume that the CBM is located 2 eV below the vacuum level. In fact, according to literature (e.g., the mentioned PhD thesis of Stefan Fölsch, Tsay and Lin, Surf. Sci. 603 (2009) and Steurer et al., Appl. Phys. Lett. 104 (2014)), the CBM of NaCl should be roughly aligned with the vacuum level. This is what we assume also in our current work. We nonetheless drew the CBM with a slight offset to the vacuum level to graphically distinguish these two. We now adapted the figure and placed the CBM very close to the vacuum level.

Request 1: Add in SI a detailed explanation for how the Fig. 4 was drawn so that anyone can draw the corresponding figures in other systems in the same way.

In regard to most levels a detailed description of how we drew Fig. 4 is already provided in the Supporting Information Section 3. We have now extended this section on page 7 of the SI to also include information on the energetic positions of CBM and VBM:

"The conduction band minimum (CBM) and valence band maximum (VBM) are indicated in Fig. 4 to highlight the need to consider the NaCl's band structure for tunneling. References [15–17] suggest that the CBM is roughly aligned with the vacuum level while the band gap is at least 8 eV. The NaCl film lowers the metal work function, i.e., 4.9 eV for Cu(111) and 5.3 eV for Au(111), by about 1 eV [18,19]. Thus, the band gap ranges from around 4 – 5 eV below the electrochemical potential of the sample to around 3.5 – 4.5 eV above."

I point out an important inaccuracy in the figure.

The band gap between CBM and VBM in Fig. 4 is not scaled.

Based on my measurement, the distance between the lines of CBM and VBM are corresponding only 7.3 eV much smaller than 8.5 eV.

Request 2: Redraw the figure with correctly scaled band gap of NaCl (see the roughly corrected figure below).

We followed the referee's advice and have now modified Fig. 4 so that also the energetic position of CBM and VBM, as well as the band gap, are to scale with the rest of the figure.

With this scaled figure, I found it's difficult to believe the effective barrier heights of the NaCl barrier in Au(111) at NIR and at PIR are similar. The apparent energy barrier is very different. The apparent barrier in the PIR case is ~ 4.1 eV, which is almost twice larger than that in the NIR case (~ 2.3 eV).

From my perspective, if the authors insist the effective barrier heights for the PIR and NIR on Au(111) are similar, it is necessary to explain or discuss why the effective barrier height becomes similar despite the very different apparent barrier height.

The authors stated in the revised manuscript "However, since the different neutralization processes involve tunnel channels with different energetic alignment with respect to the substrate's states and thus with respect to CBM and VBM (see Fig. 4), it seems unlikely that the effective barrier heights result from the energetic separation of these channels to either VBM or CBM alone, but rather that both bands can contribute to the effective barrier height. Important for the qualitative understanding of the experimental observation of a nearly energy independent effective barrier height seems to be that all contributing tunneling processes occur at energies deep inside the wide band gap of NaCl." However, this does not explain the above thing nor justify not to explain it, in my opinion.

Request 3: Explain or at least discuss why the effective barrier height becomes similar despite the very different apparent barrier heights of PIR and NIR in Au(111) case.

In the following we refer to the effective barrier height as Φ_{eff} in the exponent of the distance-dependent tunnel current

$$I \propto \exp\left(-2\sqrt{2m_e\Phi_{\text{eff}}}\frac{z}{\hbar}\right)$$

It is determined by the decay of the electron's wave function inside the tunnel barrier. In vacuum, this decay is indeed only governed by the energy difference between the corresponding tunnel channel and the vacuum level, since vacuum has no band structure to be considered for evanescent wave functions in the "forbidden energy region". However, here we consider tunneling through an insulating material (NaCl). In this case, the probability for tunneling through this material is determined by its band structure, more precisely by the complex wave vector at the energy at which the tunneling process occurs. This

complex wave vector determines how the electron's wave function decays inside the insulator. In the gap, but in direct vicinity to a band, the decay of the electron's wave function scales as if it was determined by an effective barrier height given by the energetic difference to the closest lying band. However, deeper in the gap this simplified description does not hold any more, as has been shown by complex band structure calculations for several different materials. Instead, the tunneling is governed by the imaginary part of the complex wave vector at the energy of the tunnel channel that carries the tunnel current.

Thus, we do not (and one cannot) deduce an apparent barrier height just from the sketch in Fig. 4, but we determine it experimentally from the corresponding $I_{\text{sat}}(d_{\text{NaCl}})$ curve, which is comparable to $I(z)$ in 'usual' STM measurements. As described in the manuscript and in the Supporting Information Section 4, since the tunnel current through a tunnel barrier depends exponentially on the barrier thickness, the experimental value of the effective barrier height of this tunnel process can be deduced from the slope of the semi-logarithmic plot of the current (in this case I_{sat}).

Our experimentally determined values of Φ_{eff} suggest that the effective barrier height of the different neutralization processes does not vary significantly. These neutralization processes involve tunneling at different energetic positions with respect to e.g. the CBM, and in fact, one purpose of Fig. 4 is to visualize exactly at: Showing that the tunnel channels are at remarkably different energies with respect to CBM and VBM. From this, and our experimentally determined Φ_{eff} , we deduce that the effective barrier height of the involved tunnel channels is almost energy independent. This can be explained by the energy dependence of the complex wave vector in NaCl: Inside the band gap, this wave vector has an imaginary component. The imaginary part of the wave vector must vanish at the band edges. Consequently, it must have extrema in the energy range of the band gap, where it does not change much with energy. While we did not find calculations of the complex band structure of NaCl, calculations for the wide-band-gap insulator MgO (with NaCl crystal structure) in fact show that deep inside the band gap the imaginary wave vector becomes nearly energy independent (Z. Bai et al., PRB 87 014114 (2013)). It behaves remarkably different from the oversimplified picture of an effective tunneling potential given by the band edges; please see fig. 3a of the work by Bai et al., where κ , the inverse decay length, is shown as a function of energy. We expect the band structure within the band gap region of NaCl to be qualitatively the same. This would explain our observation of the effective barrier height being independent of energy: At the energies of the tunnel channels that are involved in the neutralization of the molecules, the imaginary wave vector is almost independent of energy, leading to similar tunneling probabilities, and hence effective barrier heights, for the neutralization of anions and cations.

We described this 'level of sophistication' in the previous rebuttal letter, but did not change the corresponding section in the manuscript at that time. In light of the reviewer's comments and requests, we realize that the discussion of the energy-independent barrier height may indeed require this more in-depth discussion of the problem, and we have changed the corresponding section in the manuscript accordingly. However, the main point of the paper is not to provide a theoretical explanation of why the apparent barrier heights are similar or how they can be theoretically derived, but to provide quantitative values for the charge state lifetimes and the experimentally determined effective barrier heights. In short, the scope of the paper is to report our experimental results and summarize our (unexpected) findings, which is: a) similar effective barrier heights deep in the band gap of NaCl, almost independent of the exact energetic position of the tunneling channel within the band gap and b) reduced lifetimes related to shorter tunneling paths for tunneling processes that involve the interface state. Fig 4 serves as a visualization of

these points, showing, as the referee acknowledges, very different energetic alignments of the tunneling channels and also showing which channels couple through the interface state. It is not meant as an explanation of the quantitative effective barrier heights that we measure.

Changes in the manuscript: We changed the section in which we discuss the similar slopes of the $\ln(I_{\text{sat}}(d_{\text{NaCl}}))$ curves and the effective barrier height (page 7 and 8). The changes are highlighted in grey.

“We hypothesize that the similar m result from the effective NaCl barrier height Φ_{NaCl} being determined by the relative energetic position of the tunnel channels with respect to the band structure of the NaCl film. It has been shown that for tunnel processes involving states that are energetically near the VBM, hole tunneling, for which the tunnel barrier height is given with respect to the VBM and thus decreases with decreasing carrier energy, can dominate [39]. Especially for the cation (Fig. 4b, d), the energetic difference of one of the tunnel channels to the VBM becomes comparable to the energetic difference of tunnel channels to the CBM and vacuum level in other cases, which could explain its similarly small m compared to the anion (Fig. 4a, c). However, since the different neutralization processes involve tunnel channels with different energetic alignment with respect to the substrate’s states and thus with respect to CBM and VBM (see Fig. 4), it seems unlikely that the effective barrier heights result from the energetic separation of these channels to either VBM or CBM alone. Instead, the similar m suggest that the effective barrier height for the neutralization of the ionic molecules is nearly energy independent for the different processes studied in this work. In fact, tunnelling through a solid is correctly described by a complex band structure [40–42]: inside the band gap the wave vectors become imaginary and the wave functions decay exponentially into the bulk. Such imaginary wave vectors adequately describe the tunnelling and thereby translate into an effective tunnelling barrier. Complex band structure calculations for the wide-band-gap insulator MgO suggest that deep inside the band gap the imaginary wave vector becomes nearly independent of energy [40–42]. We expect the complex band structure of NaCl to be qualitatively similar, which would explain our observation of m being nearly independent of energy. Note that, in direct vicinity to CBM or VBM, we expect the effective barrier height to be smaller and exhibit a stronger energy dependence compared to deep in the band gap, as observed for doubly charged Cl vacancies [31]. The potential landscape being influenced by the different transient charge states of the molecule as the initial state of the tunneling process through NaCl, may further influence the effective barrier heights.”

In summary, the challenging experiment was conducted with the highest level of technique and setup. The results are new, surprising and impressive, which is clearly different from the vacuum tunneling barrier. Their hypothesis is simple and reasonable but is not well supported. I am not convinced that the results are reasonably explained. Therefore, I do not recommend the publication of the current form of the manuscript in Nature communications.

We thank the reviewer again for their assessment of our results, and hope that with the now more detailed and sophisticated description and discussion, we can convince them of our hypothesis. We agree with the referee that the experimental results are surprising and were also not what we had expected.

Sincerely,

Katharina Kaiser
on behalf of all authors

Reviewer comments, fourth round

Reviewer #3 (Remarks to the Author):

4th Review of "Charge-state lifetimes of single molecules on ultrathin insulating films" by Kaiser et al.

I appreciate that the authors sincerely responded to my comments. With the correction of Fig. 4 and the addition of further discussions, the manuscript presents the authors' hypothesis and explanation to the readers in a more understandable manner. Although I am not convinced that the results are well explained, this work contains important and new experimental results that should be widely discussed. Therefore, now I can recommend the publication of this manuscript.